



# Improved workflow for customized ICESat-2 ATL06 elevations captures seasonal mountain snow depths at sub-kilometer scale

Karina Zikan[1], Ellyn M. Enderlin[1], Hans-Peter Marshall[1], and Shad O'Neel[2]

[1]Department of Geosciences, Boise State University
[2]Cold Regions Research and Engineering Laboratory, United States Army Corps of Engineers

**Correspondence:** Karina Zikan (karinazikan@u.boisestate.edu)

**Abstract.** Mountain snowpacks are a critical global water resource, providing freshwater to more than a billion people worldwide. Estimates of snow distribution across mountain watersheds are dependent on sparse in situ observations and spatiotemporally limited aerial surveys. Models expand snow depth estimation coverage, but are also limited by the amount of snow depth measurements available for data assimilation. Satellite remote sensing provides larger spatial and temporal coverage of observations. Recent studies have shown that NASA's ICESat-2 satellite can resolve a mountain snow signal by comparing snow covered ICESat-2 returns to an independently collected snow-free digital terrain model (DTM). However, ICESat-2 snow depth uncertainties are estimated to be approximately on the order of snow depths in mountainous terrain, limiting their use. In this study we present a refined methodology for calculating ICESat-2 snow depth to minimize uncertainty in mountain environments and identify terrain with the most potential for ICESat-2 observation. We calculate snow depth by differencing a customized ICESat-2 product (ATL06_SR) and 1 m resolution DTMs of four mountainous sites in central and southern Idaho. Accuracy of ICESat-2 snow depths is assessed at 100 m - 5000 m scales through comparison with in situ automatic weather station (AWS) data and six airborne lidar snow depth surveys. By co-registering to minimize geolocation errors, correcting for slope-related biases, and filtering negative snow depth observations, we find ICESat-2 snow depth uncertainties as low as 0.2 m and $R^2$ correlation as high as 0.9 when compared to in situ observations. The median ICESat-2 snow depth over 100 m-1000 m lengths accurately captures seasonal snow depth variability and spatial patterns in snow distribution, including elevation-controlled orographic patterns. ICESat-2 snow depths are most accurate in regions where the majority of the terrain has slopes < 20° and typical winter snow depths > 0.5 m. Over 50% of Idaho's snow-covered area within this moderate slope, deep snowpack range and other mountain regions likely have sizable portions that meet these criteria. Thus by using available high-resolution snow-free DTMs, ICESat-2 applications can be dramatically expanded to provide valuable snow depth timeseries for observation or data assimilation.

## 1 Introduction

Mountain snowpacks, a key part of the global water towers, provide critical freshwater resources for over 1.9 billion people around the world (Barnett et al., 2005; Immerzeel et al., 2020; Sturm et al., 2017). Knowing the spatial and temporal distribution of snow mass is necessary to predict runoff and water availability in the melt season (Freudiger et al., 2017). Snow accumulation



processes such as orographic precipitation and wind redistribution affect the spatial distribution of snow, creating potentially

meters of snow depth variability at the slope scale (meters to hundreds of meters), mountain ridge scale (hundreds to thousands

of meters), and mountain range scale (kilometers to thousands of kilometers) (Deems et al., 2006; Elder et al., 1991; Mott et al.,

2018; Winstral and Marks, 2014). Given the high spatial variability of snow, the 2017 Decadal survey calls for 1–4 km global

snow water equivalent (SWE) resolution and ~100 m SWE resolution in the mountains to meet snow observation needs for

water management (National Academies of Sciences, Engineering, and Medicine (U.S.), 2018). SWE is calculated from snow

depth and density; currently neither snow depth nor snow density observations are available at the scale and resolutions called

for in the 2017 Decadal survey. Remote sensing is needed to meet global snow observation needs and will require integration

of many remote sensing techniques with models and ground observations (Dozier et al., 2016; Durand et al., 2021).

In this paper we calculate snow depth measurements from NASA's Ice, Cloud, and Land Elevation Satellite-2 (ICESat-2) —

a satellite lidar system. Lidar has been very successful in mapping snow depths with high spatial resolution (Deems et al., 2013;

Hopkinson et al., 2004). Unlike other optical techniques, lidar can measure the ground surface under vegetation cover (Deems

et al., 2013; Neuenschwander et al., 2020); but, like all optical approaches, lidar requires clear sky conditions. Terrestrial and

airborne lidar are used to map snow depth at sub-meter resolutions over large geographic areas, however airborne and terrestrial

lidar are still spatially and temporally limited by expense, logistics, and weather conditions (Deems et al., 2013). With repeat

global orbits, satellite lidar can help address the need for accurate and widely distributed snow depth measurements.

Although ICESat-2 was not specifically designed for snow depth retrievals, ICESat-2 can resolve a snow signal and when

used as a model constraint successfully improves estimated snow depths (Mazzolini et al., 2025). Two approaches for measur-

ing snow depth with ICESat-2 have been developed: a comparative differencing method for static terrain (Besso et al., 2024;

Chen et al., 2025; Deschamps-Berger et al., 2023; Enderlin et al., 2022; Fair et al., 2025; Hu et al., 2022a); and a multiple scat-

tering method for spatio-temporally evolving terrain (e.g., sea ice) (Hu et al., 2022b; Lu et al., 2022). In this paper we use the

comparative differencing method, where snow depth is calculated by comparing snow-on elevations from an ICESat-2 product

(e.g, ATL08 or ATL06) to an independent high-resolution snow-free digital terrain model (DTM). The comparative–difference

method has been used for snow depth mapping with airborne lidar for decades (Deems et al., 2013); conceptually its applica-

tion to ICESat-2 snow depth-mapping is the same but is complicated by ICESat-2 beam spreading, noisy photon returns, and

resolution and scale differences between ICESat-2 and reference DTMs. Much of the mid-latitude snowpack covers mountain

watersheds where steep terrain and dense vegetation complicate ICESat-2 elevation data, resulting in snow depth errors that

can be on the same order as the snow depths themselves (Besso et al., 2024; Chen et al., 2025; Deschamps-Berger et al., 2023;

Enderlin et al., 2022). Surface roughness and slope increase the spread of photon returns within the ~11 m ICESat-2 beam

footprint (Smith et al., 2019). ICESat-2 elevations and derived snow depths become less precise and increasingly negatively

biased with increased slope (Chen et al., 2025; Deschamps-Berger et al., 2023; Enderlin et al., 2022; Smith et al., 2019). Ad-

ditionally, comparing ICESat-2 data to an independently collected snow-free DTM introduces additional geolocation errors

(Enderlin et al., 2022; Hugonnet et al., 2022; Nuth and Kääb, 2011). In order to effectively use ICESat-2 to estimate snow

depths in the rugged regions that are the least constrained by observations but where the most snow accumulates there is a need

to refine ICESat-2 snow depth retrievals to minimize the aforementioned errors.



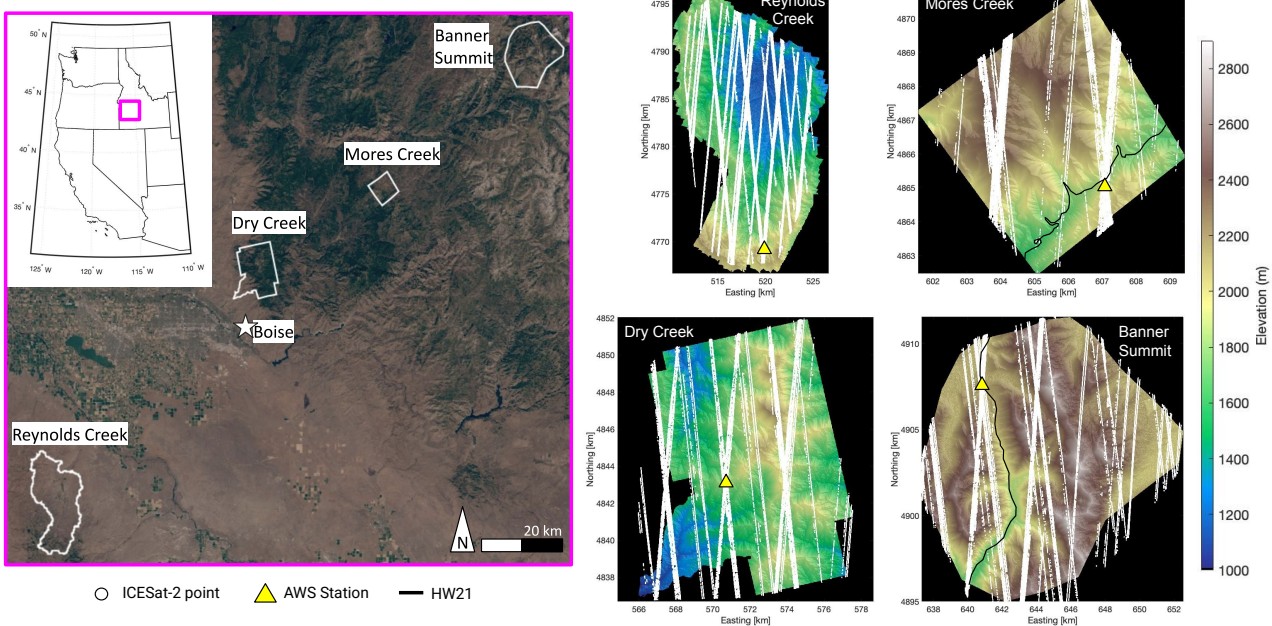

**Figure 1.** (Left) Sentinel-2 image (courtesy of European Space Agency, Copernicus) of southern Idaho with AOIs for each study site. The inset map of the Western United States shows the area within the pink square. (Right) All ICESat-2 ATL06_SR data points from 2018–2024 and AWSs overlaid on the snow-free reference DTMs of Reynolds Creek, Banner Summit, Mores Creek, and Dry Creek.

Using data from four mountainous sites in central and southern Idaho (Fig. 1), we assess the accuracy of ICESat-2 snow depths at slope to mountain ridge scales through comparison with in situ weather station data and six airborne lidar snow depth surveys. Based on analysis of ICESat-2 snow depth uncertainties in variable terrain, we identify terrain and scales at which ICESat-2 snow depth measurements have the most potential. We present a workflow for calculating ICESat-2 snow depths that can be applied in shallow to moderately sloped terrain where a high resolution snow-free DTM is available, such as mountains

covered by the USGS 3D Elevation Program (3DEP) or the Swiss national DTM (swissALTI3D), expanding the utility of ICESat-2 for snow depth observations and data assimilation.



## 2 Background and data

### 2.1 ICESat-2

ICESat-2 provides precise laser altimetry from the Advanced Topographic Laser Altimeter System (ATLAS). ICESat-2 mea-
sures elevations along three pairs of weak and strong beam tracks. Each pair of beams has a beam footprint of ~11m (Magruder
et al., 2021), an intra-pair separation of 90 m, and an inter-pair separation of 3.3 km (Neumann et al., 2019). ICESat-2 returns
have a geolocation uncertainty of ~4.4 m (Luthcke et al., 2021). Launched in 2018, ICESat-2 has a 91 day polar orbiting cycle
but it points off-nadir outside of the polar regions, repeating reference ground tracks in mid-latitudes every 3 years, to map veg-
etation. This infrequency of repeat reference ground tracks makes it impractical to monitor snow depth on the watershed scale
in mid-latitudes by comparing snow-off and snow-on ICESat-2 crossovers. We can circumvent the spatio-temporal restrictions
caused by infrequent mid-latitude ICESat-2 crossovers by comparing ICESat-2 measurements to independently collected high-
resolution snow-free DTMs. Additionally, ICESat-2's off-pointing angle can be increased up to 10 degrees to target a point
of interest, resulting in up to roughly monthly to bi-monthly acquisitions when tasked and increasing the likelihood for clear
sky conditions and usable ground observations. We conduct our analysis using observations from four Idaho study sites for
which acquisitions were tasked for every track within 30 km on an automated weather station with in situ snow depth data from
2020–present, as described below.

The ICESat-2 team releases several standard ICESat-2 products designed for different applications. For this study we use a
hybridized data product based on two higher level ICESat-2 products: the Land Water Vegetation Elevation product (ATL08)
(Neuenschwander et al., 2023) and the Land Ice Elevation product (ATL06) (Smith et al., 2023). ATL08 uses a surface finding
algorithm to identify the ground and top of canopy within the photon signal cloud (ATL03); photons are classified as ground,
canopy, top of canopy, or noise returns (Neuenschwander et al., 2023). ATL08 parses the classified photons into 100 m segments
with 0% overlap between each segment. At Reynolds Creek Experimental Watershed (Reynolds Creek), one of our Idaho study
sites, ATL08-derived snow depths calculated using a 1 m resolution airborne lidar-derived snow-free reference DTM have an
estimated normalized median absolute deviation (NMAD) error estimate of 0.95 m (Enderlin et al., 2022). ATL06 is designed
for observing ice sheets and glaciers where there is no vegetation but differences in surface elevation must be resolved on
the order of centimeters in order to capture important variations in snow and ice volume. ATL06 calculates ground elevation
from the centerpoint of a linear fit to the ATL03 signal photons within a 40 m segment with 50% overlap between segments,
resulting in an estimate of the mean ground elevation every 20 m (Smith et al., 2019). ATL06 accounts for temporary photon
detector saturation due to strong reflectance over snowy surfaces, improving snow elevation accuracy. In rough snow-covered
terrain, comparable to our mountain study sites, ATL06's vertical accuracy is expected to exceed the aforementioned ATL08
uncertainty. However, ATL06 can have a positive elevation bias in forested terrain due to a lack of vegetation filtering. When
compared to a 15 m resolution Airborne Snow Observatory (ASO) DTM for Tuolumne Basin in California, ATL06-derived
snow depths have a NMAD of 0.60-1.16 m (Deschamps-Berger et al., 2023). In this study we make use of the strengths
of both algorithms using a hybridized data product (ATL06_SR) (Besso et al., 2024) that incorporates ATL08 vegetation
filtering into an ATL06-like product using the SlideRule Earth (Shean et al., 2025) data processing package. As applied in this



paper, ATL06_SR includes ATL08's vegetation filtering but the SlideRule atl06 function does not include the first photon bias correction, which can result in up to ~2 cm of bias, or the transit pulse shape bias, which can result in up to ~1 cm of bias. ATL06_SR derived snow depths have a Root Mean Square Error (RMSE) of 0.18 m in Tuolumne Basin compared to a 3 m resolution ASO DTM and a RMSE of 0.33 m in the Methow Valley in Washington compared to a 1 m resolution airborne
lidar DTM (Besso et al., 2024). Thus we use the hybridized ATL06_SR product to calculate snow depth because of its superior precision relative to either stand-alone product (Besso et al., 2024).

## 2.2 Sites and validation data

We evaluate ICESat-2 snow depths over four mountain study sites across southern and central Idaho: Reynolds Creek Experimental Watershed (Reynolds Creek), Banner Summit (Banner Summit), Mores Creek Summit (Mores Creek), Dry Creek
Experimental Watershed (Dry Creek). These four sites span an array of vegetation cover, elevation ranges, terrain complexity, and average snow depth. Each site has a high-resolution snow-free airborne lidar DTM raster available. These DTMs are from several campaigns: the NASA SnowEx 2020–21 Time Series campaign (Adebisi et al., 2022), available at NSIDC; an Airborne Snow Observatory (ASO) campaign (Ilangakoon et al., 2014), available through Boise State University; and combined University of Idaho and FEMA campaigns (Quantum Spatial, 2018), available through the Idaho Lidar Consortium. For each site, the
area of interest (AOI) was defined to match the extent of the site's snow-free DTM. To validate the ICESat-2 snow depths, we use in situ snow depths from automatic weather stations (AWS) within each study site. Each site has a SNOwpack TELemetry Network (SNOTEL) AWS; some sites have additional AWSs described below. SNOTEL is a network of automatic weather stations (AWS) in snow-dominated watersheds across the western United States, operated by the USDA NRCS Snow Survey. The SNOTEL stations provide hourly measurements of snow depth with a precision of 1.3 cm. Additionally, at Mores Creek,
multiple 1 m resolution airborne lidar surveys of snow depth across the Mores Creek AOI were flown over the 2022–24 winter seasons. Each site is described in more detail below.

### 2.2.1 Reynolds Creek Experimental Watershed

Reynolds Creek Experimental Watershed (Reynolds Creek) is a 239 km$^2$ semi-arid rangeland watershed in the Owyhee Mountains in Southwestern Idaho (Fig. 1). At lower elevations, the vegetation is predominantly low semi-arid shrubs, such as sage-
brush, and cultivated land; in the mid-elevations, juniper is prevalent, and at higher elevations there are copses of Douglas fir and aspens (Seyfried et al., 2018). Elevations in the watershed range from 1100 m to 2245 m and span the typical rain–snow transition elevation. Since the 1960s, the Reynolds Creek rain-to-snow transition zone has increased from ~1300-1600 m to 1669–2013 m (Seyfried et al., 2018). In 1960, the USDA established the site as a snow hydrology research basin administered by the USDA Agricultural Research Service (ARS) Northwest Watershed Research Center (NWRC). The USDA operates four
clustered snow observation AWS in Reynolds Creek at elevations of 2054 m asl to 2097 m asl (Center). The four AWSs were operational for different, but overlapping time periods. The data for all four AWS was concatenated and nominally geolocated to the average location of the four AWSs creating a complete snow depth record for the observation period. The AWSs are within 40 m of each other, therefore within a single 40 m ATL06_SR segment, thus any snow depth variation between





AWS should be within the snow depth variation captured by the ICESat-2 segment. A 1 m resolution snow-free lidar DTM of
Reynolds Creek was collected by NASA's Jet Propulsion Laboratory (JPL) Airborne Snow Observatory (ASO) in August 2014
(Ilangakoon et al., 2014) and can be accessed through the Idaho lidar Consortium (https://www.idaholidar.org/).

### 2.2.2   Banner Summit

The Banner Summit study area is a 183 km$^2$ section of alpine terrain in the Sawtooth Mountains in Central Idaho (Fig. 1).
Banner Summit is predominantly covered in conifer forest. Banner Summit elevations range from 2061 m to 2810 m and are
typically above the rain–snow transition line from December to May, but span a wide range of snow conditions. The Banner
Summit SNOTEL station (Station #312) is installed at 2146 m. A 0.5 m resolution snow-free DTM of Banner Summit was
collected by Quantum Spatial, Inc in September 2021 as part of the NASA SnowEx campaign (Adebisi et al., 2022). This study
area includes the most avalanche-prone highway in Idaho, HW21, through an area known as "avalanche alley". The majority
of the site burned in the 2024 Wapiti wildfire, however the data used in this study was collected before the fire.

### 2.2.3   Mores Creek Summit

Mores Creek Summit (Mores Creek) is a 38 km$^2$ alpine study site in the Boise Mountains 15 km north of Idaho City in Western
Idaho (Fig. 1). Mores Creek is predominantly covered in conifer forest; roughly 50% of the site burned in the 2016 Pioneer
wildfire, leaving standing dead trees and a rapidly evolving understory. Mores Creek elevations range from 1790 m to 2369
m and are typically above the rain–snow transition line from December to April. The Mores Creek SNOTEL station (Station
#637) is installed at 1859 m. A 0.5 m resolution snow-free DTM of Mores Creek was collected by Quantum Spatial, Inc in
September 2021 as part of the NASA SnowEx campaign (Adebisi et al., 2022). Helicopter-based airborne lidar snow depth
surveys at 1 m horizontal resolution and 10 cm depth uncertainty over the 38 km$^2$ Mores Creek AOI were collected during the
2022–24 winter seasons (Ciafone et al., 2024). Surveys were collected during the snow season (November – April); one survey
in the 2021–22 season, two surveys in the 2022–23 season, and three surveys in the 2023–24 season were collected within
2 days of an ICESat-2 overpass (Table 1). Of the 6 surveys within 2 days of an ICESat-2 overpass, three have no snow free
segments due to either complete snow cover of the observed terrain or cloud contamination of snow-free terrain.

### 2.2.4   Dry Creek Experimental Watershed

The Dry Creek Experimental Watershed (Dry Creek) study site is a 126 km$^2$ area of the Boise Front Range in western Idaho
including the 27 km$^2$ Dry Creek research basin operated by the hydrology group at Boise State University (McNamara et al.,
2018). In this water-limited environment, vegetation in the lower watershed is predominately a sparse sagebrush-steppe ecosys-
tem with dense riparian vegetation in the valley bottoms. Mixed conifer forest is the dominant land cover through the upper
watershed (McNamara et al., 2018). Elevations in the watershed range from 1160 m to 2050 m spanning the typical rain–snow
transition elevation. The Lower Deer Point weather station operated by Boise State University is installed at 1850 m (McNa-
mara et al., 2018). A snow-free lidar DTM of the Dry Creek research basin was collected in 2007 (Sciences, 2007), the upper



**Table 1.** Mores Creek helicopter survey dates and temporally nearest ICESat-2 Mores Creek overpass. Stared dates have available snow-free ATL06_SR data to perform an individual co-registration.

| Mores Creek snow depth survey date | Closest ICESat-2 Mores Creek overpass date | Interval days | Snow-free ICESat-2 segments | Snow-on ICESat-2 segments |
|---|---|---|---|---|
| 17 February 2022 | 15 February 2022 ★ | 2 | 40 | 344 |
| 16 March 2023 | 17 March 2023 | - 1 | 0 | 874 |
| 15 November 2023 | 13 November 2023 ★ | - 2 | 133 | 150 |
| 15 January 2024 | 16 January 2024 | 1 | 0 | 485 |
| 13 February 2024 | 14 February 2024 | 1 | 0 | 146 |
| 16 April 2024 | 18 April 2024 ★ | 2 | 26 | 810 |

section of the Dry Creek site was surveyed in 2019 by FEMA (Quantum Spatial, 2018). These two 1 m resolution DTM were combined to create the Dry Creek snow-free reference DTM used in this study. Both DTMs can be accessed through the Idaho Lidar Consortium (https://www.idaholidar.org/). We chose to use these DTMs instead of the existing SnowEx lidar snow-free survey of Dry Creek due to their larger spatial coverage of the area.

## 3   Methods

We calculate ICESat-2 snow depths in four main steps: (1) pre-process ICESat-2 elevation data (h_IS2) from the study site; (2) co-register the ICESat-2 elevation data with the snow-free reference DTM; (3) extract snow-free reference elevations (h_DTM) and terrain parameters (slope, aspect) within each ICESat-2 segment; and (4) calculate ICESat-2 snow depth (d_IS2) and snow-free elevation residuals (h_residual) using

$$d\_IS2 = h\_IS2(snow) - h\_DTM(ground) \tag{1}$$

$$h\_residual = h\_IS2(ground) - h\_DTM(ground) \tag{2}$$

The snow-free ICESat-2 height residuals, h_residual, are the difference between ICESat-2 and DTM ground elevations when and where snow was not observed in near-coincident satellite imagery. In idealized conditions h_residual would be zero. The variation in h_residual is assumed to be from uncertainties in the ICESat-2 and DTM data, natural snow depth variation within the ~11 m diameter ICESat-2 laser footprint, differences in the spatial coverage of elevation returns within each ICESat-2
segment due to differences in sensor design, and geolocation offsets between the datasets. h_residual is used to calculate and correct elevation biases between the datasets and calculate the ICESat-2 precision and snow-free accuracy. ICESat-2 snow depth, d_IS2, is the difference between ICESat-2 and DTM elevations when and where snow is observed in satellite imagery.



After accuracy analyses, ICESat-2 snow depths are validated against in situ AWS snow depths and the Mores Creek airborne lidar snow depth surveys.

As we developed the workflow presented here, we tested several other changes to these methods that we either found to be impractical or have little to no impact on ICESat-2 snow depth precision or accuracy. These workflow variations are described in Appendix A.

### 3.1    Pre-process ICESat-2

We use ATL06_SR (Besso et al., 2024) for all available ICESat-2 data acquired from October 2018 to April 2024 over our four
study sites. To calculate ATL06_SR, we apply the SlideRule ATL06 function to ATL08 ground-classified photons (as in Besso et al., 2024), using otherwise default ATL06 parameters. ATL06_SR has a segment length of 40 m with 50% overlap and a photon threshold of 10 photons. We use the ATL06_SR h_mean parameter — the elevation at the center of a linear fit of the photons within the 40 m segment, equivalent to the area-averaged elevation — as h_IS2 for our analysis. Since the ATL06_SR h_mean parameter is evaluated at the center of the ATL06_SR segment, we refer to the nominal location of the elevation data
as the centroid. The ATL06_SR and reference DTM data are reprojected to WGS84 UTM zone 11N (EPSG:32611). The native coordinate reference frame (CRS) for ICESat-2 is the WGS84 ellipsoid (EPSG: 4326). The native CRS for the snow-free reference DTMs is the North American Datum 1983 in the horizontal (EPSG: 26911) and the North American Vertical Datum 1988 (EPSG: 5103) geoid heights in the vertical.

### 3.2    Horizontal co-registration

Geolocation offsets increase uncertainty and introduce error in the interpretation of ICESat-2 elevation data compared to the DTM, especially in sloped terrain. The magnitude of the error depends on the magnitude of the geolocation offset as well as the surface slope and aspect relative to the horizontal direction of offset (Nuth and Kääb, 2011). Since the snow-free reference DTMs of each site are independent of each other, ICESat-2 and the reference DTM must be co-registered on a site by site basis. Accurate co-registration relies on stable surface matching, thus only snow-free ICESat-2 elevation data are used for co-
registration. Snow-free ICESat-2 elevations are identified using the temporally closest Sentinel-2 (courtesy of European Space Agency, Copernicus) or Landsat 8/9 (courtesy of NASA and USGS) derived snow cover map. Snow cover is identified using the Normalized Difference Snow Index (NDSI), wherein

$$NDSI = \frac{\rho_{green} - \rho_{SWIR}}{\rho_{green} + \rho_{SWIR}} \tag{3}$$

where $\rho_{green}$ and $\rho_{SWIR}$ are the surface reflectance in the green and SWIR bands respectively. $\rho_{green}$ and $\rho_{SWIR}$ are bands
B3 and B11 for Sentinel-2 imagery and bands SR_B3 and SR_B6 for Landsat 8/9 imagery (Gascoin et al., 2019; Hall et al., 1995; Riggs et al., 1994). For each study site, NDSI time series are created using all Sentinel-2 and Landsat 8/9 images covering at least 70% of the AOI and with < 10% cloud-cover in the AOI during our observation period (2018–24). Pixels with an NDSI > 0.4 are classified as snow (Gascoin et al., 2019; Riggs et al., 1994). ATL06_SR segments are assigned as snow-covered or



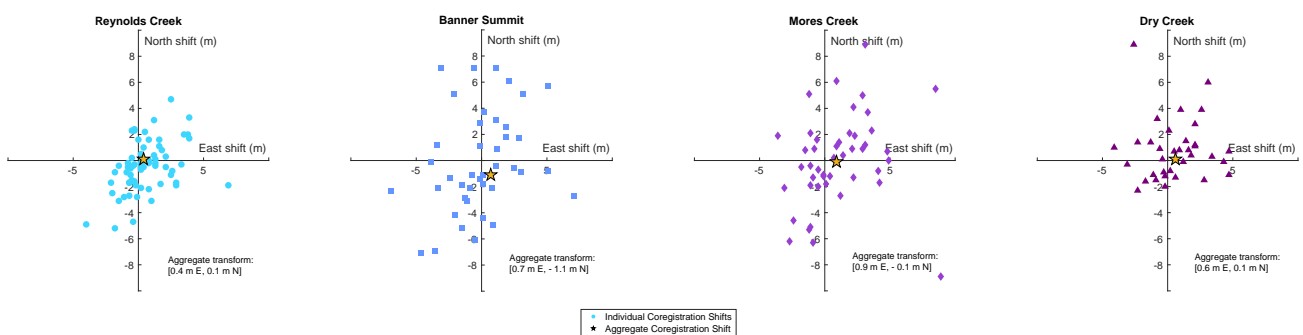

**Figure 2.** Aggregate and individual co-registration northing and easting transforms for Reynolds Creek, Banner Summit, Mores Creek, and Dry Creek. Stars show the aggregate co-registration transform. Colored markers show the individual co-registration transforms. The aggregate co-registration transform for Reynolds Creek = [0.4 m E, 0.1 m N], Banner Summit = [0.7 m E, - 1.1 m N], Mores Creek = [0.9 m E, - 0.1 m N], and Dry Creek = [0.6 m E, 0.1 m N].

snow-free based on the classification of the nearest neighbor snow cover pixel. The numbers of snow-free and snow-covered

segments for each site are in Table S1.

After identifying the snow-free ICESat-2 elevation data, ATL06_SR and the site DTM are co-registered using an iterative grid search that is similar to the iterative co-registration used in Zhang et al. 2024 to reduce errors in DTMs of the rugged Qinghai-Tibet Plateau. First a coarse (1 m increment) grid search is performed over a grid of +/- 8 m in the easting and northing directions — almost double the ICESat-2 geolocation uncertainty of ~4.4. m (Luthcke et al., 2021) — to find the

approximate location of the global minimum of the snow-free ICESat-2 elevation residual NMAD. A finer (0.1 m increment) grid search is then performed within +/- 0.9 m of the coarse grid search's minimum to identify the true global minimum of the snow-free ICESat-2 elevation residual NMAD. At each coordinate in the transform grid the mean reference elevation in each ICESat-2 segment is calculated to downsample the reference DTM to the geometry of the transformed ICESat-2 tracks before calculating the elevation residual NMAD. Additional co-registration approaches that were tested and rejected during method

development are discussed in Appendix A.

Most existing ICESat-2 snow depth studies use an aggregate co-registration approach where a single co-registration transform is calculated from all snow-free ICESat-2 elevations and applied to all ICESat-2 tracks (Deschamps-Berger et al., 2023; Enderlin et al., 2022). We test both an aggregate co-registration approach using all snow-free ATL06_SR segments and an individual co-registration for each ICESat-2 overpass using all snow-free ATL06_SR segments from the overpass (Fig. 2). For

comparison with the results of Enderlin et al. 2022, who estimated a near-zero offset, and Besso et al. 2024, who applied no horizontal transform, we also test a "no co-registration" approach wherein the ICESat-2 elevation residuals and ICESat-2 snow depths are calculated without any horizontal co-registration transform. Final reported ICESat-2 snow depths are calculated using aggregate horizontal co-registration but discussion of other co-registrations is included for comparison.



### 3.3 Calculate reference elevation, slope, and aspect

After horizontal co-registration, we calculate elevation, aspect, and slope metrics from the reference DTM for each ATL06_SR segment. The reference elevation for each ATL06_SR point is the mean elevation of the DTM within the corresponding ATL06_SR segment area (a 11 m by 40 m rectangle, oriented along the ICESat-2 track). The along-track slope and aspect are calculated from a plane fit to the reference DTM surface within the corresponding ATL06_SR segment area. Although the terrain metrics are extracted from within the entire ATL06_SR segment area, the terrain metrics are nominally assigned the

same segment centroid coordinates as the ATL06_SR elevation to facilitate analysis.

### 3.4 Vertical co-registration and snow depth estimation

Vertical co-registration is calculated using snow-free conditions and applied to both snow-free and snow-covered data. Since the snow-free ICESat-2 elevation residuals are not normally distributed, we solve for any vertical bias and the progressive negative slope bias that has been previously identified in ICESat-2 data in mountain regions (Deschamps-Berger et al., 2023;

Enderlin et al., 2022; Liu et al., 2021; Smith et al., 2019) using the median snow-free elevation bias from slopes of 0° to 40° in 5° steps. Slopes > 40° are excluded to remove outliers. As this progressive slope bias is best described by a quadratic function (Enderlin et al., 2022), we fit a quadratic curve to the median snow-free ICESat-2 elevation residuals relative to slope to solve for site-specific vertical elevation corrections. The vertical bias is assumed to persist in snow-covered conditions, thus the quadratic snow-free vertical bias correction is applied to the snow-covered ICESat-2 returns to minimize biases in

snow depth. Transects of 40 m ATL06_SR snow depth estimates are compared to near-coincident airborne lidar snow depth surveys of Mores Creek. To evaluate the scale at which ICESat-2 best captures snow depth patterns we calculate the median ICESat-2 snow depth within various lengths, herein we refer to this length as the smoothing length. The AWS snow depth time series at each site are compared to the median ICESat-2 snow depths within 100 m, 500 m, 1000 m, and 5000 m of the AWS. The median ICESat-2 snow depth is compared to the median airborne snow depth at 100 m, 500 m, 1000 m, and 5000

m smoothing lengths. Segment scale spatial variability is evaluated using the NMAD to limit the impact of the heavy-tailed residual distribution while the median snow depth temporal variability is evaluated using the root mean square error (RMSE) as the residual distribution is not heavy-tailed.

### 3.5 Mores Creek helicopter snow surveys data processing

The Mores Creek helicopter snow depth surveys are co-registered to the snow-free Mores Creek DTM using HW 21 as a

control feature as elevation along the plowed road is stable across seasons (Fig. 1) (Hoppinen et al., 2023). The aggregate and individual co-registration transforms calculated for the snow-free Mores Creek DTM are applied to the Mores Creek ICESat-2 data to co-register the ICESat-2 data to the Mores Creek helicopter snow depth surveys. For three of the six Mores Creek helicopter survey dates there are insufficient data to calculate an individual co-registration transform (Table 1) so those dates are excluded from the individual co-registration analysis.





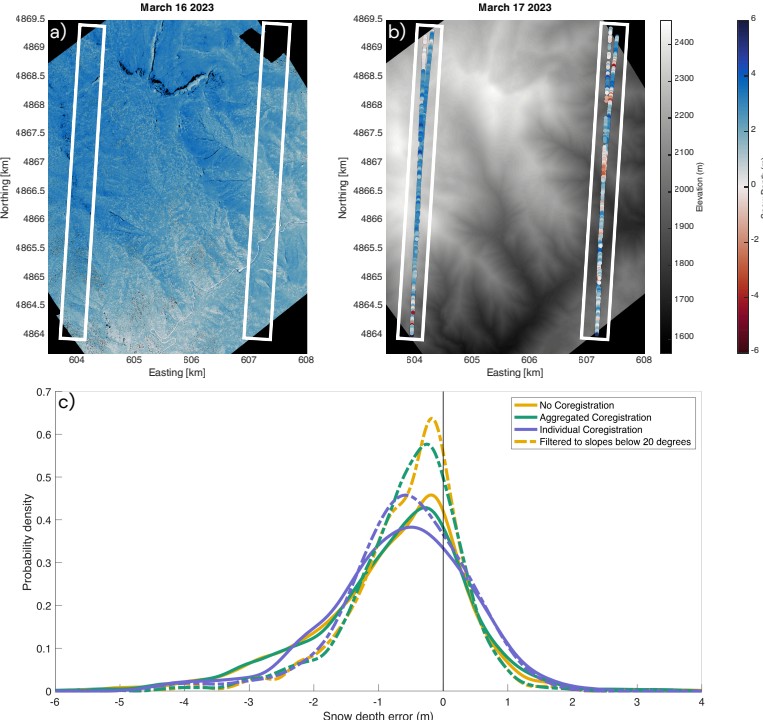

**Figure 3.** a) 16 March 2023 airborne lidar snow depth map. b) 17 March 2023 ICESat-2 snow depths calculated using aggregate co-registration overlaid on the Mores Creek snow free reference DTM. Both ICESat-2 snow depths and airborne snow depths are plotted with the same colormap. Blue indicates positive snow depths and red is negative snow depths. White boxes in both (a) and (b) highlight the location of the ICESat-2 tracks from 17 March 2023. c) Probability distribution of snow depth error for all paired survey dates calculated without co-registration, using aggregate co-registration, and using individual co-registration before and after filtering the segments to slopes < 20°.

To compare ICESat-2 snow depths to the helicopter lidar snow depth surveys of Mores Creek, snow depth profiles are calculated along the corresponding ICESat-2 overpass at the ATL06_SR resolution, creating synthetic ICESat-2 tracks (Fig. 3). Reference helicopter snow depths are calculated as the average snow depth of the helicopter snow depth survey within the ATL06_SR segment and nominally assigned the centroid coordinates for the corresponding segment.

## 4   Results

Since the focus of this paper is optimization of ICESat-2 processing to estimate snow depths in rugged terrain, we first present the impact of co-registration and slope corrections on ICESat-2 snow depth accuracy and precision then present a detailed comparison of ICESat-2 snow depths to independent snow observations.



**Table 2.** Snow-free elevation residual NMADs for Reynolds Creek, Banner Summit, Mores Creek, and Dry Creek and Mores Creek ICESat-2 snow depth error NMAD, Median, and $R^2$ compared to the Mores Creek airborne snow depth surveys calculated using no co-registration, aggregate co-registration, and individual co-registration transforms.

| Snow-free ICESat-2 residual NMAD (m) | | | | | | | |
|---|---|---|---|---|---|---|---|
| Co-registration approach | Reynolds Creek | Banner Summit | Mores Creek | Dry Creek | NMAD of snow depth error (m): Mores Creek | Median of snow depth error (m): Mores Creek | Snow depth $R^2$: Mores Creek |
| None | 0.41 | 0.89 | 1.09 | 1.21 | 0.64 | - 0.27 | 0.47 |
| Aggregate | 0.41 | 0.88 | 1.08 | 1.22 | 0.68 | - 0.28 | 0.47 |
| Individual | 0.37 | 0.98 | 1.10 | 1.07 | 0.74 | - 0.33 | 0.00 |

## 4.1 Aggregate vs Individual Co-registration Transforms

To facilitate comparison with previous analyses of ICESat-2 snow depth mapping, we calculate bias (i.e., median error) and uncertainty (i.e., NMAD of error) without co-registration, using aggregate co-registration transforms, and using individual co-registration transforms. The magnitude of the individual co-registration transforms reach up to 8.9 m in a single direction (Fig. 2), which is more than double the expected precision (4.4 m) of ICESat-2 geolocation (Luthcke et al., 2021). In contrast, the magnitude of the aggregate co-registration transforms reach a maximum of 1.1 m in a single direction (Fig. 2). The directions of the individual co-registration transforms are generally NE or SW, with no clear pattern for ascending or descending ICESat-2 tracks. The snow-free elevation residual NMAD is < 0.5 m at Reynolds Creek, ~1 m at Banner Summit and Mores Creek, and ~1.2 m at Dry Creek regardless of co-registration approach. Individual co-registration results in inconsistent changes of ~10 cm to ICESat-2 precision across sites (table 2). To assess the influence of co-registration approach on snow depth estimates, we compare snow depth from ICESat-2 and synthetic ICESat-2 tracks calculated from near-coincident Mores Creek airborne lidar snow surveys. We find that ICESat-2 snow depth has a negative bias of ~0.6 m and uncertainty of ~1 m regardless of co-registration approach (Fig. 3, Table 2). Additionally, the coefficient of determination ($R^2$) between the ICESat-2 and airborne lidar snow depths indicates stronger correlation between the variables without co-registration or using aggregate co-registration of ICESat-2 ($R^2$ ~0.5) relative to individual co-registration ($R^2$ ~0).

We attribute the unreliability of the individual co-registration to the abundance of snow cover in the study site: for several winter ICESat-2 overpasses the site's AOI is almost completely snow covered, resulting in < 100 segments for individual co-registration and a high likelihood that segments identified as snow-free are partially snow-covered. The aggregate co-registration uses ~7000 - 100000 snow-free segments for co-registration depending on site while the individual co-registration uses an average of ~100 - 2000 snow-free segments per overpass date depending on site for co-registration. Thus the individual co-registration has orders of magnitude fewer data points available for co-registration, likely skewing transforms for some of the individual tracks.





**Table 3.** Uncertainty and accuracy statistics before and after applying a site-specific quadratic slope correction.

| Site | | Reynolds Creek | | | Banner Summit | | | Mores Creek | | | Dry Creek | | |
|---|---|---|---|---|---|---|---|---|---|---|---|---|---|
| Before or after slope correction | | N | Before | After | N | Before | After | N | Before | After | N | Before | After |
| **Median** | **All slopes** | 160652 | -0.78 | 0.02 | 18228 | -14.35 | -0.02 | 7317 | -1.07 | 0.01 | 20113 | -1.29 | 0.03 |
| **ICESat-2** | **< 10°** | 80125 | -0.67 | 0.02 | 4529 | -13.93 | -0.06 | 859 | -0.58 | 0.03 | 700 | -0.66 | 0.01 |
| **Elevation** | **>10°, < 20°** | 62911 | -1.02 | 0.04 | 5828 | -14.37 | 0.01 | 3672 | -0.96 | 0.01 | 4950 | -0.98 | 0.06 |
| **Residual (m)** | **> 20°** | 17616 | -1.56 | 0.01 | 7873 | -15.29 | 0.07 | 2788 | -1.65 | 0 | 14485 | -1.53 | 0.02 |
| | **All slopes** | 160652 | 0.56 | 0.41 | 18228 | 1.26 | 0.88 | 7317 | 1.17 | 1.08 | 20113 | 1.3 | 1.22 |
| **ICESat-2** | **< 10°** | 80125 | 0.25 | 0.24 | 4529 | 0.29 | 0.36 | 859 | 0.55 | 0.57 | 700 | 0.39 | 0.37 |
| **Elevation** | **>10°, < 20°** | 62911 | 0.71 | 0.68 | 5828 | 0.88 | 0.83 | 3672 | 1.03 | 1.01 | 4950 | 0.87 | 0.85 |
| **NMAD (m** | **> 20°** | 17616 | 1.29 | 1.27 | 7873 | 1.75 | 1.64 | 2788 | 1.49 | 1.47 | 14485 | 1.48 | 1.45 |

## 4.2 Slope-induced errors

Uncertainty and bias in ICESat-2 elevation residuals, and therefore snow depths, both increase with slope (Table 3, Fig. 4). Bias is effectively minimized through the application of a quadratic slope-dependent adjustment but the bias adjustment has a slope- and site-dependent impact on uncertainty. At all slopes the quadratic slope-dependent adjustment removes vertical bias but the slope-dependent bias adjustment has little impact on uncertainty. The magnitude of uncertainty increases with slope at all sites (Fig. 4). We define shallow, moderate, and steep slopes as < 10°, 10 - 20°, > 20° respectively, based on the distribution of slopes across our sites (Fig. 2a). For shallow slopes site-specific uncertainties range from ~0.2 - 0.6 m, for moderate slopes uncertainties range from ~0.7 - 1.0 m, and for steep slopes uncertainties range from ~1.3 - 1.8 m both before and after applying the slope-dependent bias adjustment (Table 3).

## 4.3 Snow depth comparisons at stations

To assess the optimal spatial smoothing length of ICESat-2 snow depths in rugged terrain, median ICESat-2 snow depth is calculated within 100 m, 500 m, 1000 m, and 5000 m of the AWS within each site. The daily in situ AWS snow depths are used as ground truth. Across all sites 20%–50% of ICESat-2 snow depth segments are negative, and thus clearly inaccurate. At all sites, removing negative ICESat-2 snow depth segments improves median ICESat-2 snow depth agreement with AWS snow depth across all smoothing lengths; the RMSE for all ICESat-2 snow depths at each site ranges from 0.3 m to 1.9 m, shrinking to 0.2 m to 0.8 m when negative ICESat-2 snow depth segments are excluded (Table 4). If negative ICESat-2 snow depths are not excluded, the median ICESat-2 snow depth systematically underestimates AWS snow depth by an average of 0.54 m across all sites and smoothing lengths; with negative ICESat-2 snow depths excluded, median ICESat-2 snow depth systematically underestimates AWS snow depth by an average of 0.22 m (Fig. 5). Because of this improvement to ICESat-2 snow depth accuracy and precision, negative ICESat-2 snow depth segments are excluded from the ICESat-2 snow depth results presented below.



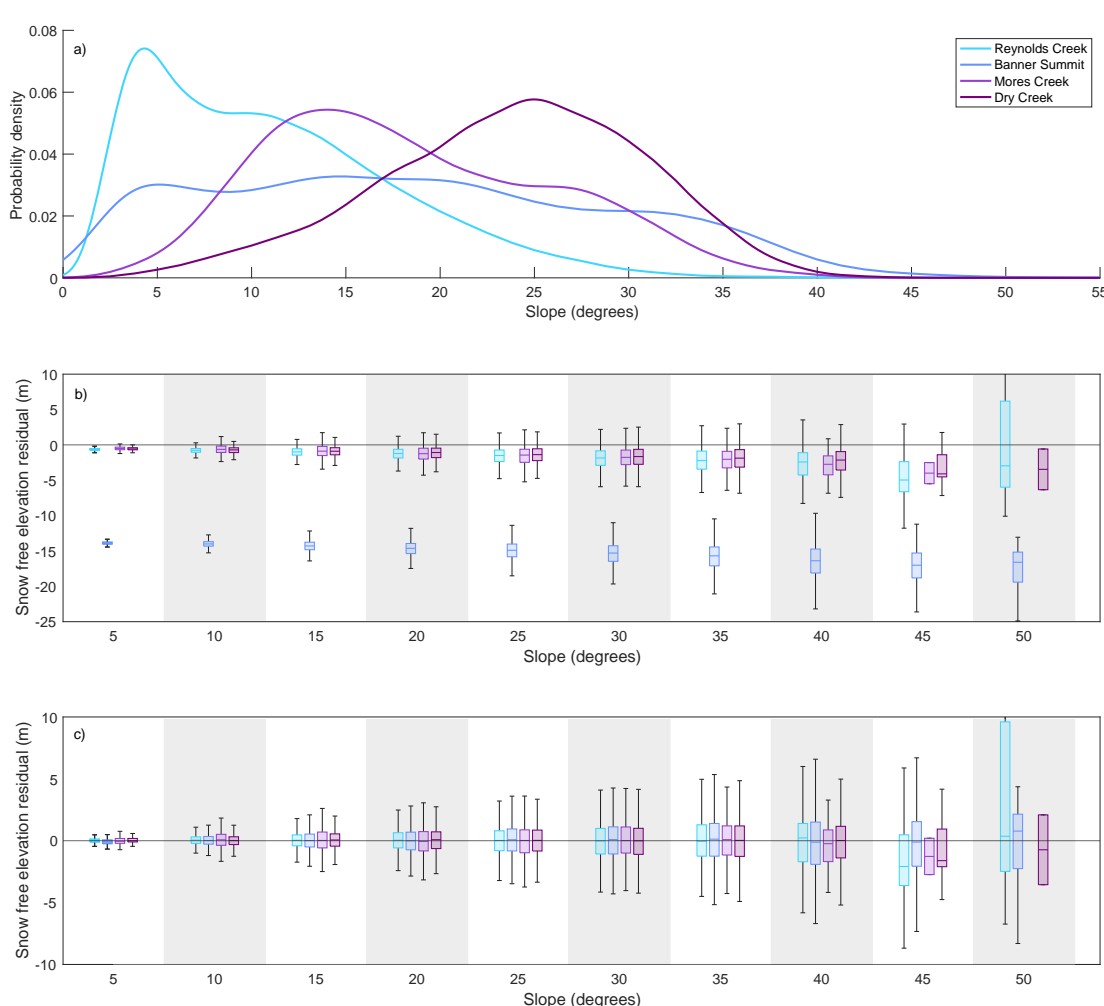

**Figure 4.** a) Normalized distribution of slopes within each study site. b & c) Snow-free ICESat-2 elevation residual binned by slope for each site before (b) and after (c) quadratic slope correction. The residuals before slope correction [b] are shown on a scale from -25 m to 10 m to show the Banner Summit residuals which have a large vertical offset. The corrected residuals [c] are shown on scale from -10 m to 10 m.





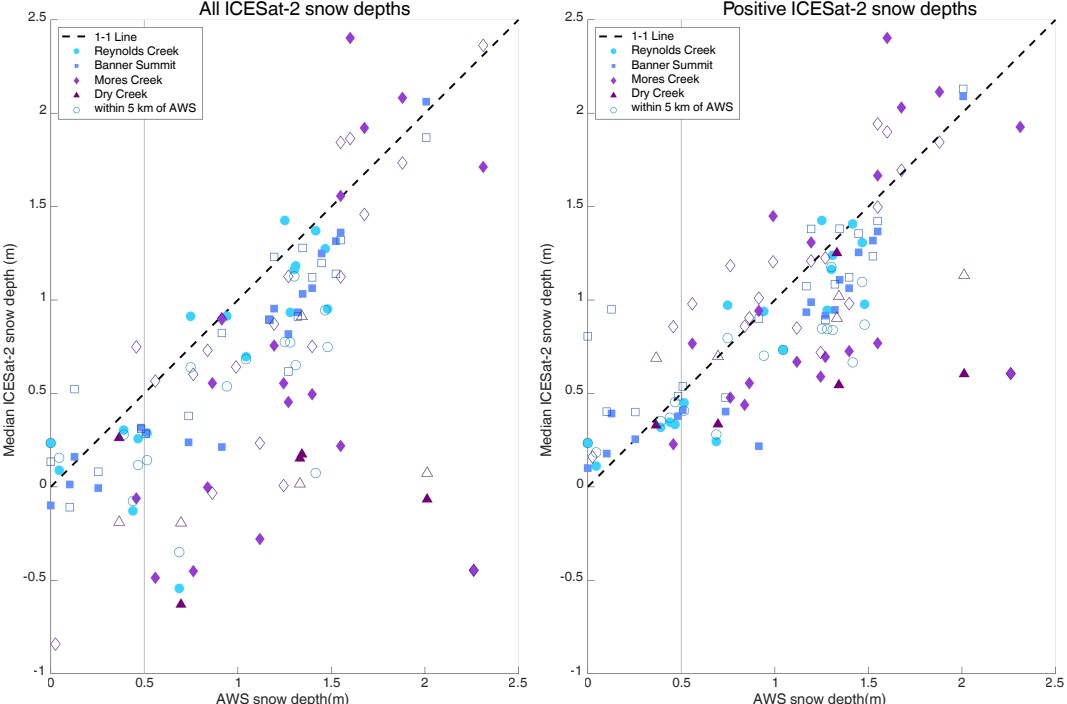

**Figure 5.** AWS snow depth vs median ICESat-2 snow depth with (left) and without (right) negative ICESat-2 snow depth values. The dotted line is the 1-to-1 line. Marker shape and color distinguish sites. Filled markers indicate the median ICESat-2 snow depth within 500 m of the AWS and empty markers indicate the median ICESat-2 snow depth within 5 km of the AWS.

With negative ICESat-2 snow depth segments removed, the median ICESat-2 snow depth generally captures temporal accumulation and ablation patterns over the snow season at the Reynolds Creek, Banner Summit, and Mores Creek AWSs (Fig. 5, Fig. 6, Table 4). We define the best smoothing length as the length which minimizes snow depth uncertainty while maximizing correlation with the AWS snow depth. At Reynolds Creek Banner Summit, and Mores Creek, the AWS snow depth is best captured by ICESat-2 at a 500 m smoothing scale where the snow depth uncertainty reaches its minimum and the coefficient of determination ($R^2$) for a linear regression fit to the ICESat-2 snow depths relative to AWS snow depths reaches its maximum for the full time period at each site (Table 4). At Reynolds Creek $R^2$ is also near or at maximum for both the accumulation season (October-March; 0.85) and ablation season (April-September; 0.80) at the 500 m smoothing scale. At Banner Summit, snow depth $R^2$ is ~0.9 for both the accumulation season and the ablation season at the 500 m smoothing scale. Unlike the other sites, Dry Creek's ICESat-2 snow depth uncertainty and correlation with AWS observation improve as the smoothing scale increases, however these statistics are all calculated from $\leq 5$ observations at all lengths and are not considered robust.

Conversely, when capturing the spatial distribution of snow across a watershed larger smoothing scales have lower uncertainty and greater correlation to observed snow depths. Comparing ICESat-2 snow depths to the Mores Creek Airborne surveys



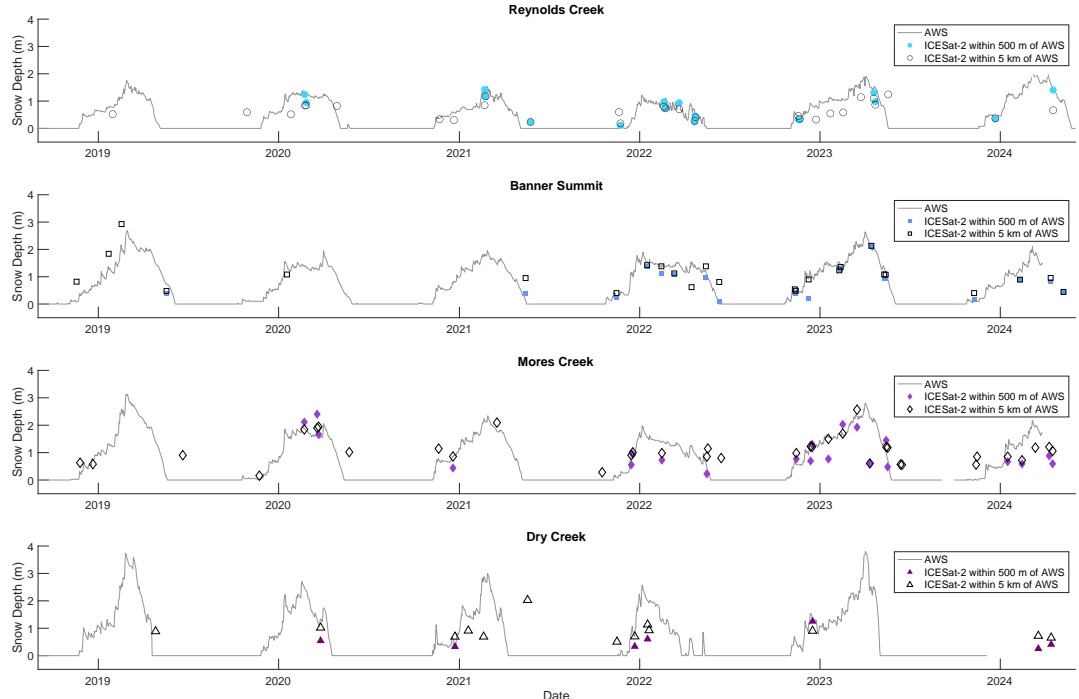

**Figure 6.** Time series of median daily in situ AWS snow depth and median ICESat-2 snow depth with negative ICESat-2 snow depth segments removed for each ICESat-2 overpass. Marker shape and color distinguish sites as in Fig. 5. Filled markers indicate the median ICESat-2 snow depth within 500 m of the AWS and empty markers indicate the median ICESat-2 snow depth within 5 km of the AWS.

ICESat-2 snow depth uncertainty and correlation with observation improve as the smoothing scale increases (RMSE = 0.38 m,
$R^2$ = 0.81 at smoothing scale = 5000 m and filtered to slopes < 20°). The ICESat-2 snow depth bias remains ~-0.4 m to ~-0.6 m at all smoothing scales. To assess ICESat-2's ability to capture terrain dependent variations in snow depth, we calculate the snow depth elevation gradient at 100 m elevation increments using ICESat-2 snow depths within 5 km of the AWS for all sites and airborne lidar for Mores Creek. To reduce outliers, the median ICESat-2 snow depth is not calculated for elevation bins with fewer than 30 ICESat-2 observations. In February and March, the months of near-peak snow depth at the AWS, the
ICESat-2 snow depth gradient with elevation is 0.11 m per 100 m of elevation gain at Reynolds Creek, 0.12 m at Banner Summit, and 0.18 m at Dry Creek (Fig. 7). At Mores Creek, the ICESat-2 and airborne lidar snow depth gradients with elevation are 0.23 m and 0.24 m per 100 m of elevation gain, respectively.





**Table 4.** Snow depth $R^2$ , RMSE, and number of snow depth observations for ICESat-2 snow depths relative to AWS snow depths within 100 m, 500 m, 1000 m, and 5000 m of the AWS at each site. Underlined numbers are the highest $R^2$ and lowest RMSE at each study site for all ICESat-2 snow depths (left columns) and positive-only ICESat-2 snow depths (right columns). Statistics are not reported for Dry Creek due to limited data ( $\leq$ 5 dates with ICESat-2 observations).

| Site | | Reynolds Creek | | Banner Summit | | Mores Creek | |
|---|---|---|---|---|---|---|---|
| ICESat-2 Snow Depths | | All | Positive only | All | Positive only | All | Positive only |
| Snow Depth $R^2$ | Within 100 m | 0.37 | 0.58 | 0.92 | 0.91 | 0.21 | 0.17 |
| | Within 500 m | 0.64 | 0.85 | 0.91 | 0.92 | 0.41 | 0.6 |
| | Within 1 km | 0.44 | 0.66 | 0.91 | 0.79 | 0.32 | 0.43 |
| | Within 5 km | 0.01 | 0.78 | 0.66 | 0.55 | 0.24 | 0.46 |
| Snow Depth RMSE (m) | Within 100 m | 0.5 | 0.36 | 0.29 | 0.25 | 0.78 | 0.57 |
| | Within 500 m | 0.39 | 0.21 | 0.31 | 0.22 | 1.16 | 0.43 |
| | Within 1 km | 0.51 | 0.3 | 0.31 | 0.3 | 0.99 | 0.49 |
| | Within 5 km | 0.96 | 0.35 | 0.43 | 0.4 | 0.72 | 0.45 |
| Number of dates with ICESat-2 observations | Within 100 m | 16 | 16 | 15 | 15 | 14 | 14 |
| | Within 500 m | 17 | 16 | 18 | 15 | 20 | 14 |
| | Within 1 km | 18 | 18 | 21 | 21 | 21 | 20 |
| | Within 5 km | 31 | 18 | 23 | 21 | 34 | 21 |

# 5 Discussion

We find that the uncertainty in snow free ICESat-2 elevation residuals varies from ~0.4 m to 1.2 m depending on site. However,
uncertainties in snow depths may be less than inferred from a comparison of snow-free elevations; the NMAD for the near-coincident Mores Creek airborne snow depth surveys and ICESat-2 snow-depths is ~0.5 m less than the NMAD for snow-free Morse Creek terrain. ICESat-2 can capture the temporal variations in snow depth measured by AWS with an RMSE as low as 0.2 m and $R^2$ as high as 0.9, at relatively flat study plots. ICESat-2 also captures spatial distribution patterns of ICESat-2 when smoothed over sufficiently large scales. Although there is still the potential for methodological improvements, ICESat-2
is useful for mapping variations in snow depths over 100 m to kilometer scales in regions where there are slopes < 20° and snow depths exceed the approximately half-meter uncertainty, as discussed in more detail below.

## 5.1 Individual vs aggregate co-registration

Careful co-registration is a critical step to reduce geolocation errors in the ICESat-2 snow depth signal, yet it remains a challenge in complex snow-covered terrain that lacks stable controls. For our final ICESat-2 snow depth analysis we apply an
aggregate co-registration transform calculated from all the snow-free ICESat-2 observations to maximize stable (snow-free) observations. The orders of magnitude more data available for co-registration when the tracks are aggregated help minimize





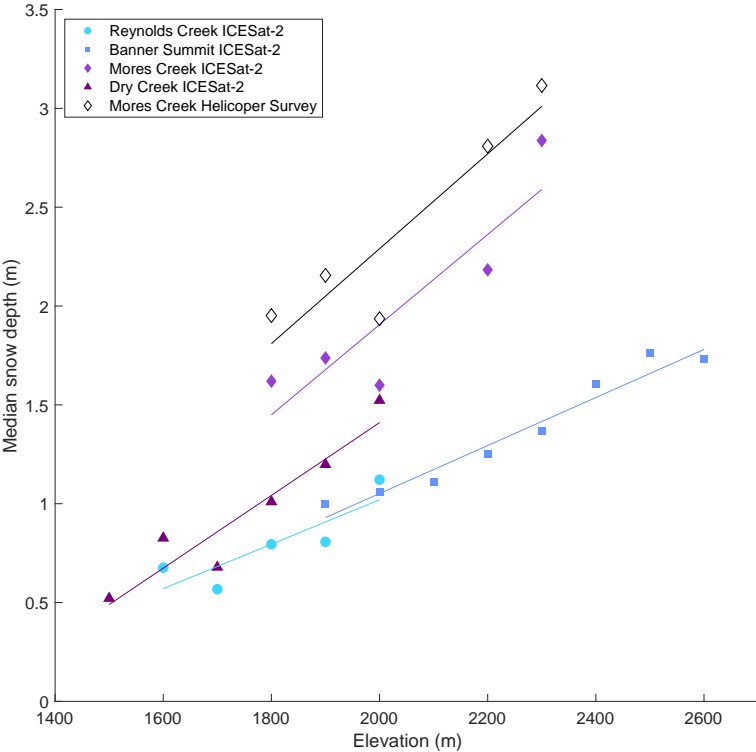

**Figure 7.** Median February and March snow depths within 5 km of the AWS by 100 m elevation increments. Lines show a linear fit to the median snow depth. The Mores Creek ICESat-2 data has been filtering to only overpass dates near-coincident with the airborne snow depth surveys.

the impact of ICESat-2 noise, misidentified snow cover, and suboptimal terrain for cross-correlating distinct features with high accuracy.

There is no inherent reason why the co-registration of ICESat-2 tracks relative to a reference DTM should be constant over time, yet we found little change in uncertainty and accuracy of ICESat-2 snow depth estimates when applying no co-registration transform and a time-invariant co-registration transform while applying time-variant co-registration transform performed slightly worse than either other co-registration approach in most cases (Table 2, Fig. 3). There is large variability in estimated individuals co-registration transforms which can exceed ICESat-2 geolocation uncertainty (4.4 m; Luthcke et al. 2021, Fig. 2). Most concerningly, the application of a time-variant co-registration transform resulted in no correlation between ICESat-2 snow depths and precise independent snow depth estimates (Table 2). The poor performance of individual co-registration transforms is likely due to sparse snow-free winter terrain. Snow coverage obscures stable terrain and when <10% of the region of interest is stable the accuracy of co-registration decreases with the percent of stable terrain (Nuth and



Kääb, 2011). At the sites in this study the individual ICESat-2 overpasses frequently have very few snow-free segments (< 10% of segments).The impact of snow cover on snow-free ICESat-2 segment availability can clearly be seen at Mores Creek where only 50% of the paired ICESat-2 and Mores Creek airborne lidar dates have any snow-free ICESat-2 returns at all with which to perform an individual co-registration. There are indications of the impact of snow-free segments in the individual co-registration transforms at each site: at Reynolds Creek and Dry Creek, the two sites that span the typical rain–snow transition line and thus often contain both snow-free and snow-covered ICESat-2 segments, there is less variability within the individual co-registration transforms than for the mostly snow-covered Banner Summit and Mores Creek (Fig. 2). It is possible that, provided a sufficient amount of snow-free observations, an individual co-registration transform calculated for each ICESat-2 overpass may be more accurate than an aggregate co-registration transform. Individual co-registration may be particularly important for tasked study sites, such as these, because the tasked off-nadir pointing angle can vary by up to 10° and uncertainty increases with deviation from an orthogonal incidence angle (Smith et al., 2019). However, for an individual co-registration viable for snow observation, a large portion of the reference DTM must be snow-free in the winter.

## 5.2   Snow depth

ICESat-2 snow depth mapping can capture temporal and spatial snow distribution patterns but its efficacy is dependent on surface slope. In this study we use multiple datasets for comparison error assessment: snow-free ICESat-2 elevations to snow-free reference DTMs; median ICESat-2 snow depths to AWS snow depths; and ICESat-2 snow depths to airborne lidar-based snow surveys. Each has its strengths and weaknesses. The snow-free ICESat-2 comparison to the reference DTM has the most data and is downsampled to the same spatial coverage and resolution as snow-free ICESat-2, but it is only a proxy for snow covered uncertainty. Snow depth uncertainties are generally less than snow-free uncertainties, likely because ICESat-2 returns a stronger signal from the smoother, brighter, and vegetation-free snow surface. The spatial coverage difference between ICESat-2 (a track of points across an area) versus the AWSs (a point observation) likely contributes to differences in the median ICESat-2 snow depth and AWS snow depth. Although airborne snow depths are only available for six dates at Mores Creek, they are likely a better snow depth validation dataset than the AWS snow depth as the airborne survey is downsampled to the same spatial resolution and coverage as ICESat-2. Our comparison with these data indicate that ICESat-2 systematically underestimates observed snow depth but accurately captures orographic distribution patterns across the 38 km$^2$ area (Table 2, Fig. 3, Fig. 7). Though Mores Creek has a higher snow depth uncertainty than Reynolds Creek or Banner Summit due to its steep terrain, uncertainties in ICESat-2 snow depths compared to the airborne snow depth surveys are ~0.5 m when spatially smoothed and ~0.7 m for individual segments, indicating that ICESat-2 is capable of capturing snow depth variability in shallow-to-moderate slope terrain.

Slope strongly influences uncertainty and bias in snow-free ICESat-2 elevations, and therefore snow depths (Table 3). As the majority of our in situ snow depth observations are point observations, we use snow-free ICESat-2 elevation residuals to gain insight on the impact of slope on uncertainty. We find snow-free uncertainties of ~0.2 - 0.6 m for slopes < 10° while the snow-free uncertainty increases by ~1 m or more when slopes are > 20°. While it is difficult to directly assess snow depth uncertainty with respect to slope, the magnitude of snow depth uncertainty at each study site follows this pattern given the



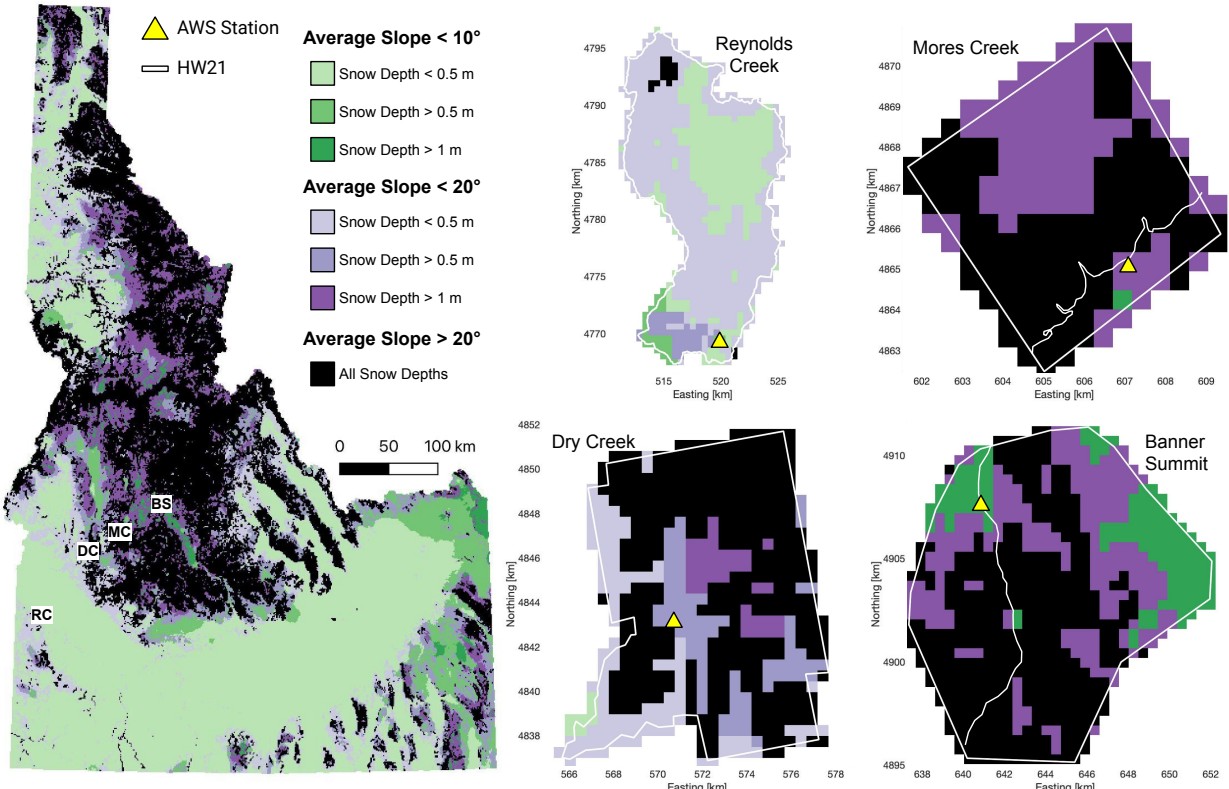

**Figure 8.** Map of slope zones and snow depths at 1 km scale across Idaho. Cutouts show slope zones and snow depths at 500 m scale for Reynolds Creek, Banner Summit, Mores Creek, and Dry Creek. Slopes are calculated from the 30m scale NASADEM (NASA JPL, 2020) and then averaged over 500 km. Snow depths are average early March SNODAS snow depth (National Operational Hydrologic Remote Sensing Center, 2004) from 2020–24.

distributions of slope at each site; Reynolds Creek and Banner Summit — the sites with the largest proportion of slopes < 20° — have the lowest ICESat-2 snow depth uncertainty and highest $R^2$ across the sites, highlighting the potential utility of ICESat-2 for snow depth observations in shallow-sloped locations (Fig. 8). When compared to the Mores Creek airborne snow
depth surveys, ICESat-2 snow depth agreement with the airborne observations is distinctly improved by removing observations over >20° slope terrain (Fig. 3). At Dry Creek, where the majority of slopes are > 20° and ICESat-2 observations are space, snow depths are unreliable.





Steep slopes and abundant, mixed vegetation influence actual snow depths through their influence on wind redistribution and the surface energy balance (Dharmadasa et al., 2023; Mott et al., 2018; Winstral and Marks, 2014) and influence apparent

snow depths through their impact on lidar returns to the sensor. Low bushy vegetation can influence both real and apparent snow depths. Low, dense bushes can be misclassified as ground in the snow free reference DTM and can create the illusion of negative snow depth (Gould et al., 2013; Hopkinson et al., 2004; May et al., 2025; Tinkham et al., 2011, 2014). Bushes are found in all our sites and likely bias the median reference elevation when misidentified as ground, resulting in apparent negative snow depths. The Reynolds Creek and Banner Summit AWSs are surrounded by rolling and relatively open terrain, likely

contributing to both site's small snow depth uncertainty (~0.2 - 0.4 m). The higher snow depth uncertainty at Mores Creek and Dry Creek (~0.5 - 0.8 m) is likely due to the steep and vegetated terrain surrounding the AWS at each site. Removing negative ICESat-2 snow depths before calculating the median ICESat-2 snow depth reduces bias and uncertainty of the ICESat-2 snow depth estimates, but is sensitive to the vertical co-registration. If negative values are not removed, the median ICESat-2 snow depth systematically underestimates AWS snow depths (Fig. 5). However the data suggests that shallow (< 0.5 m) snow depths

may be slightly over-estimated by the median ICESat-2 snow depth when negative ICESat-2 snow depths are removed (Fig. 5). The reductions in bias and uncertainty that result from the removal of negative snow depths are greater than if the data are filtered to remove snow depths over slopes > 20°, indicating that at least a portion of the erroneous snow depths are caused by vegetation effects.

Spatial smoothing of ICESat-2 snow depths can reduce uncertainty, though the optimal smoothing length depends on terrain

and application because actual snow depths and errors vary with slope and vegetation. A larger smoothing length will reduce random errors but may span a wide range of snow depths, including snow-free terrain. A smaller smoothing length may span a smaller range of snow depths, making the estimate more directly comparable to a point in situ observation, but will decrease the number of ICESat-2 segment observations within the smoothing length thus increasing the impact of random errors. We find that when trying to capture the temporal snow depth patterns at a AWS (a point observation) snow depth is best captured by

ICESat-2 at a smoothing scale of 500 m. The 500 m smoothing length contains less terrain variability than longer lengths while still containing a reasonable number of ICESat-2 segment observations (average 13 ICESat-2 segments) as opposed to the 100 m smoothing length (average 3 ICESat-2 segments). Conversely by comparing ICESat-2 to the Mores Creek airborne snow depth surveys we find that ICESat-2 snow depth uncertainties and correlation with observed spatial snow distribution patterns increases with smoothing scale, apparently stabilizing at a 1000 m smoothing scale. To calculate the snow depth elevation

gradient we were able to maintain larger numbers (> 30) of segment observations at a 100 m scale by grouping the data into elevation zones across the watershed rather than 100 m spatial regions.

## 5.3 Mapping Applicability for ICESat-2 Snow Depth Measurement

ICESat-2 snow depth mapping is promising, capturing spatial and temporal distribution patterns at scales of 100–1000 m; however, ICESat-2 snow depths are not reliable in all conditions. Based on the results for our study sites, we recommend

using ICESat-2 snow depth mapping in regions with a majority of slopes < 20° and for snow depths > 0.5 m. For example at Mores Creek (Fig. 7a), when ICESat-2 returns from steep slopes (> 20°) are filtered, snow depth uncertainty is ~0.3 - 0.7



m. Although ICESat-2 snow depth uncertainties can be as low as 0.2 m where the majority of slopes are < 10°, based on the relative abundance of shallow-sloped regions at our sites and across Idaho more broadly, restricting ICESat-2 application to such shallow-sloped regions will limit their usefulness. Our work reveals that slope introduces a progressive negative bias and increasing uncertainty. Calculating and applying a site specific slope correction removes the progressive bias and reduces uncertainty up to 30% so ICESat-2 can be used for snow depth observation in more complex terrain with appropriate spatial averaging. However, we are unable to remove the slope dependence of uncertainties as the slope impact on uncertainty is grounded in the size of the ICESat-2 laser footprint. Even with these corrections, sites like Dry Creek where steep slopes (majority > 20°) and bushy terrain increase snow depth uncertainty to 1.5 m and snow depth error to 1.2 m are ill-suited for ICESat-2 observation.

As a reference, we demonstrate where ICESat-2 is expected to yield useful snow depth estimates in Idaho based on slope and the snow depth relative to an estimated uncertainty of ~0.5 m. We use the 30 m NASADEM (NASA JPL, 2020) to estimate slope and the average March SNODAS model output (National Operational Hydrologic Remote Sensing Center, 2004) to estimate peak snow depths (Fig. 8). The state is first classified by slope at 10° increments, then the slope classes are parsed based on their approximate snow depth (Table 3). Although uncertainties in SNODAS and the NASADEM are large, the classified map of Idaho demonstrates why Mores Creek is a reasonable site for ICESat-2 snow depth mapping but Dry Creek is not: though Mores Creek is predominantly steep terrain, the snowpack is deep, whereas Dry Creek is steep and has a shallow snowpack. In contrast, Banner Summit, where we see the lowest ICESat-2 snow uncertainty and highest correlation with in situ snow depth, has a deep snowpack and contains shallow-sloped regions that are ideal for accurate ICESat-2 snow depth estimates. In addition to these slope and snow depth thresholds, we recommend that future applications of ICESat-2 consider both the density and type of vegetation in the study region ](Neuenschwander et al., 2020) to maximize the likelihood of canopy penetration and ground identification.

Here we use the NASADEM for visualization of slope ranges, however we do not recommend NASADEM for ICESat-2 snow depth mapping due to its coarse resolution (30 m) and potential vegetation effects. Fortunately, there are publicly available high resolution (< 5 m) DTMs of many snow dependent regions. For example, 3DEP has publicly available 1 m resolution DTM coverage for over 70% of the Western United States and is working toward 100% coverage. Utilizing these snow-free DTMs in conjunction with ICESat-2 data has the potential to drastically expand snow observation in mid-latitudes.

## 6    Conclusions

Detailed snow depth observation of key watersheds is critical to understanding present and future water availability as the changing climate strains snow water resources. Satellite remote sensing provides a path to meet global snow observation needs in mountain environments where observations are limited. NASA's ICESat-2 satellite lidar altimeter shows promising terrain-dependent potential for snow depth profiling; however, high ICESat-2 uncertainties from terrain, vegetation, and geolocation offsets complicate returns. In this study, we refine ICESat-2 snow depth measurement workflows for mountain environments,



validate spatial and temporal snow distribution patterns against in situ and airborne snow depth observations, and identify
terrain where ICESat-2 snow depth mapping has the most potential.

ICESat-2 snow depths were calculated by differencing snow-covered ATL06_SR (a customized ICESat-2 ATL06-like product that implements photon classification from the ATL08 product, Besso et al. 2024) elevations from snow-free reference DTMs. ICESat-2 snow depths were calculated for four sagebrush-steppe and mixed-conifer alpine study sites of varying terrain complexity in central and southern Idaho: Reynolds Creek, Banner Summit, Mores Creek, and Dry Creek. The median
ICESat-2 snow depths calculated at spatial smoothing lengths of 100 m, 500 m, 1000 m, and 5000 m were validated against in situ AWS snow depths in each site. At Mores Creek, ICESat-2 snow depths are additionally compared to six airborne lidar snow depth surveys of the site.

We find that ICESat-2 snow depth mapping is most successful in areas where the majority of slopes are < 20° and snow depths are > 0.5 m (Fig 5, Fig 6). At sites where the majority of slopes are < 20° (Reynolds Creek, Banner Summit, and Mores
Creek) the median ICESat-2 snow depth uncertainty is ~0.2 - 0.4 m with a $R^2$ correlation of ~0.6 - 0.9 to the AWS snow depth at a smoothing length of 500 m (Table 4, Fig. 5a). The airborne surveys suggest ICESat-2 best captures spatial snow distribution patterns at a 1000 m scale, however terrain controlled patterns can be resolved at scales as small as 100 m by grouping the data into zones across the watershed. As previously identified, there is a progressive negative slope bias in the ICESat-2 data (Fig. 4, Deschamps-Berger et al. 2023; Enderlin et al. 2022). ICESat-2 snow-free elevation uncertainty increases by ~1 m when
slopes are > 20°, compared to slopes < 10° (Fig. 4, Table 3). A quadratic slope correction removes the bias and improves the uncertainty except when slopes are < 10° (Table 3). ICESat-2 snow depth uncertainty and correlation generally improve as the smoothing length decreases. Removing negative ICESat-2 snow depths before calculating the median ICESat-2 snow depth improves uncertainty and correlation at all smoothing scales (Table 4, Fig. 5). The improvement is most pronounced at sites with few slopes < 10° (Mores Creek).
While ICESat-2 snow depth mapping is not reliable in all environments, large portions of Idaho fall within the moderate slope, deep snowpack terrain perimeters optimal for ICESat-2 observation (Fig. 8). Similarly large portions of other mountainous regions likely also contain terrain where ICESat-2 snow depth observation is reasonable. Utilizing available high resolution DTMs of these areas, such as those from USGS 3DEP, in conjunction with ICESat-2 data can drastically expand spatial and temporal snow observation coverage. While ICESat-2 cannot solely describe the snowpack, ICESat-2 snow depth measure-
ments weave into the larger array of snow observation techniques and modeling efforts creating a more complete and robust understanding of our seasonal snow reservoirs.

*Code and data availability.*  The code used to calculate ICESat-2 snow depths is available at https://github.com/CryoGARS-Glaciology/ICESat2-AlpineSnow or https://zenodo.org/records/17101987

The code used to map snow cover is available at https://github.com/CryoGARS-Glaciology/ndsi-snow-maps

The ICESat-2 snow depth datasets are currently being reviewed and will be available at https://scholarworks.boisestate.edu/cryogars_snow_data/



## Appendix A:  Method Development

ICESat-2 snow depth mapping is a relatively new method, thus optimal workflows are still being developed. As we developed the workflow presented herein we tested several other changes to co-registration, ICESat-2 processing, and snow-free reference elevation estimation that we either found to be impractical or have little to no impact on ICESat-2 snow depth precision or accuracy. To help guide future improvements to ICESat-2 snow depth mapping we briefly discuss the workflow variations we explored and areas where there are potential opportunities for improvement.

There is no inherent reason why the co-registration for ICESat-2 tracks relative to a reference DTM should be constant over time, yet we found that application of a time-invariant co-registration transform resulted in ICESat-2 snow depths that were closest to precise independent snow depth estimates (Table 2, Fig. 3). We attribute the larger uncertainty in individually co-registered ICESat-2 snow depths to imprecise co-registration due to sparse snow-free pixel in a given overpass and misclassification of snow-free segments using near-coincident Sentinel-2 and Landsat 8/9 NDSI maps. The lack of snow-free segments available for co-registration could be addressed by including snow-free winter terrain in the AOI. We attempted to include winter snow-free terrain where we could but were limited by available high-resolution DTM coverage. Temporal separation between ICESat-2 and Landsat/Sentinel-2 observation, cloud masking, and mixed cover pixels can all cause misclassification of ATL06_SR segments. To minimize contamination of the snow-covered and snow-free segments the thresholds for snow-covered and snow-free segments could be raised.

In our preliminary method development we tested two separate horizontal co-registration approaches in addition to the iterative grid search we use in the finalized workflow: the Nuth and Kääb (2011) co-registration approach (as used in Deschamps-Berger et al. 2023), and a gradient descent (as used in Enderlin et al. 2022). All co-registrations used the snow-free ICESat-2 residual NMAD as the minimization parameter. We were only able to perform a Nuth and Kääb (2011) co-registration in Reynolds Creek because the smaller study sites did not have sufficient data for robust co-registration using this approach. Both the Nuth and Kääb (2011) co-registration and the gradient descent co-registration identified horizontal transforms on the order of 1e-4 m while the grid search identified larger horizontal transforms of up to 1.1 m using aggregated ICESat-2 data and up to 8.9 m using individual ICESat-2 data. The larger transforms have a smaller snow-free ICESat-2 residual NMAD then the near-zero transforms found using the Nuth and Kääb (2011) co-registration and the gradient descent. We decided to use the iterative grid search for its reliability, however by adjusting the gradient descent initial guess and resolution a gradient descent should capture the global minimum found using the interactive grid search. For a larger AOI with more available data and complex terrain the Nuth and Kääb (2011) co-registration may be appropriate.

The accuracy of the ICESat-2 snow depths are dependent on the snow-free reference elevations calculated from the snow-free reference DTM. In addition to using the mean DTM elevation within each ATL06_SR segment as the reference ground elevation we tested linear interpolation of the DTM to the ICESat-2 segment centroids (similar to Besso et al. 2024 and Deschamps-Berger et al. 2023) and weighted mean elevation approaches. For the weighted mean approach, across-track Gaussian-weighting was applied in an attempt to mimic the distribution of photons due to beam spreading. For our study



sites, snow depth calculated using the mean DTM elevation as the snow-free reference elevation most closely matches the validation snow depths.

*Author contributions.* EE conceptualized the project. KZ and EE developed the data processing codes. KZ processed the data and performed the analysis with supervision from EE, SO, and HM. KZ prepared the manuscript with contributions from all co-authors.

*Competing interests.* The authors declare that they have no conflict of interest.

*Acknowledgements.* This work was supported by the National Aeronautics and Space Administration Established Program to Stimulate Competitive Research grant 80NSSC20M0222 and the NSF GRFP 2023353206. We would like to thank the members of the CryoGARS-Glaciology lab for their feedback and assistance downloading and processing ICESat-2 tracks; we are grateful to Rainey Aberle for providing the snow cover mapping code. Additionally, we thank Kate Hemingway for offering their expertise as a proofreader.



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
