# Peer review of "Improved workflow for customized ICESat-2 ATL06 elevations captures seasonal mountain snow depths at sub-kilometer scale"

_EGUsphere, 2025_

## Referee Comment (RC1)

**Review of**

**Improved workflow for customized ICESat-2 ATL06 elevations captures seasonal mountain snow depths at sub-kilometer scale**

**Zikan et al. 2025-4813**

**General comment**

This article presents a workflow using ICESat-2 elevations and reference snow-off digital terrain models to map snow depth in four mountainous study sites in the western US over a total of 586 km². The accuracy and precision of the retrieval is evaluated with reference snow depth from automatic weather station and from airborne lidar. The benefits of several steps of the workflow is tested (slope related correction, negative values filtering, coregistration strategy). An empirical definition of areas where snow depth surveyed with ICESat-2 is likely to be informative and valuable is proposed and used to estimate the coresponding area in the Idaho state (US).

The article reads well. The methodology and results are well presented. Some results are insightfull, such as the impact of the coregistration strategy and of the slope related correction. However, the quantification of the improvement brought by this specific workflow in comparison with previous studies is lacking. This work builds upon workflows presented in previous work which it aims to improve (Enderlin et al., 2022; Besso et al., 2024). I would expect a part of the discussion to be dedicated to a comparison of uncertainty and precision metrics to highlight the improvement brought by this workflow. Without this, it is hard to evaluate the benefits of it. A comparison with the results in Deschamps-Berger et al. (2023) and Chen et al. (2025) could also be included since they share a common general framework of differencing with external elevation models. Regarding the comparison with Besso et al. (2024), see the comment about Line 103-105.

César Deschamps-Berger

**Minor comments and suggestions**

Suggested modifications are in bold italic.

- L14 Provide the RMSE and R2 with 2 decimals precision, like in Table 4.
- **L25** Order chronologically the citations.
- **L28** Precise whom Decadal Survey it is or turn the sentence in passive mode « Goals of global snow water equivalent (SWE) at 1–4 km resolution and ~100 m SWE resolution *were set* for mountain regions to meet snow observation needs for water management (Decadal Survey...) »
- L31 «; » replace with « but » or another similar conjunction.
- L31 Maybe also state that SWE cannot be directly observed at the relevant scale and resolution?
- L34 « In this paper, » comma.
- L35 It is implied but state clearly that this refers to airborne and terrestrial lidar.
- **L37-38** no need to repeat « airborne and terrestrial Lidar » the second time?
- **L41** Some quantified values of the ICESAt-2 snow depth error found in previous studies would be welcome. See for instance, Fair et al. (2025).

- **L43** « for static terrain ... for spatio-temporally evolving terrain » This formulation got me confused and is innacurate since Lu et al. (2022) also apply their method on land masses.
- **L64** « can be applied in shallow to moderately sloped terrain » I thought that one of the conclusion of the article is that snow depth below 0.5 m are not well monitored with ICESat-2?
- **L65** « covered by the USGS 3D Elevation Program (3DEP) or the Swiss national DTM (swissALTI3D) » Move to Discussion.
- L70 Say that the high-frequency acquisition provides continuous acquisitions along-track.
- L72 « Launched in 2018, ICESat-2 has a 91 day polar orbiting cycle but it points off-nadir outside of the polar regions *to increase coverage and to map vegetation*, repeating reference ground tracks in mid-latitudes every 3 years *only*. » Suggestion.
- L74 « of repeat reference ground tracks » => « tracks »
- L75 I would turn this sentence in passive mode: « the spatio-temporal restriction *can be circumvented* ». And cite the works which did that.
- L79 I would split this sentence in two. If I understood correctly, something along: « We conduct our analysis using observations from four Idaho study sites for which *ICESat-2* acquisitions were tasked *to cover/overpass* an automated weather station within each site. *This resulted in shifting the satellites tracks within a 30 km radius around the stations and enabled the comparison of satellite and* in situ snow depth data from 2020–present, as described below »
- L81 « 2020–present » find a formulation that will remain true in a few years
- **L84** « ATL06 » as I understand the workflow, ATL06 is not used but an ATL06-like product (L100). Should it say here that this study uses a hybridized product based on ATL08 and ATL03 resulting in a ALT06-like product?
- **L89** NMAD. You may want to cite the original article as well, Höhle and Höhle (2009).
- Höhle, J., & Höhle, M. (2009). Accuracy assessment of digital elevation models by means of robust statistical methods. *ISPRS Journal of Photogrammetry and Remote Sensing*, 64(4), 398-406.
- **L91** « where there is no vegetation but *where* differences in surface elevation must be resolved on the order of centimeters in order to capture important variations in snow and ice volume »
- L95 « vertical accuracy is expected to exceed » not sure if it exceeds in absolute value ( Accuracy\_ATL06 > Accuracy\_ATL08) or in quality (ATL06 better than ATL08, Accuracy\_ATL06 < Accuracy\_ATL08)
- L100 Move (Besso et al., 2024) and (Shean et al., 2025) together at the end of the sentence.
- L101 I would avoid mentionning the « SlideRule atl06 function » which is a bit cryptic. Simply list what corrections and filters are applied or not.
- L103-105 « ATL06\_SR derived snow depths have a Root Mean Square Error (RMSE) of 0.18 m in Tuolumne Basin compared to a 3 m resolution ASO DTM and a RMSE of 0.33 m in the Methow Valley in Washington compared to a 1 m resolution airborne lidar DTM (Besso et al., 2024). » These value of RMSE are misleading as they are not the RMSE at the ATL06\_SR resolution of 20 m. This is the RMSE of the ensemble of median differences calculated for each day with ICESat-2 data. This can be verified in the code from Besso et al. (2024). For instance for the Methow Valley:

https://github.com/bessoh2/icesat2\_sr/blob/main/methow\_valley/notebooks/comparison\_to\_snotel.ipynb

In the box 56, the RMSE (0.33 m) is calculated on the object *comp\_df*, itself defined in box 54 and 55. It is filled with median differences, one for each acquisition date.

When pushing the comparison of these results with Besso et al. (2024), this code might be useful to ensure the consistency of the variables compared.

L105 « Thus, » comma.

**L111-114** That is a lot of acronyms, institutions name, and programs name. Could it be rather put in a table (acquisition date, resolution, program, provider...)? Possibly in supplement.

« Each site has a high-resolution snow-free airborne lidar DTM raster freely(?) available acquired during various campaigns (Table XX). »

These information are repeated in the study sites description. Maybe that is even enough then.

**L119** « 1.3 cm » source ?

L122 2.2.1-4 Could the basins presentation be ordered from north to south? or another geographical logic?

L127 «  $\sim$ 1300-1600 m to 1669–2013 m » keep the same precision for both ranges.

**L186** « ICESat-2 snow depth precision or accuracy » I see the point in distinguishing precision and accuracy but please provide the definition used in this article. Is one the bias, the dispersion, the combination of both...?

**L191** « a *minimal number of* photon threshold »?

**L236** « The reference elevation for each ATL06\_SR point is the mean elevation of the DTM within the corresponding ATL06\_SR segment area (a 11 m by 40 m rectangle, oriented along the ICESat-2 track) » nice effort.

L238-240 « Although the terrain metrics... » I do not understand this. Isn't it the case for ATL06 products?

**L251** « To evaluate the scale... » I am unsure how the smoothing is done on ICESat-2 elevation : along-track or within a square, circle shape? The latter would make the number of data points smoothed variable.

L271 « and precision then present » add a comma somewhere?

**L282** « *T*able 2 »

L287 The exact R2 values can be provided (0.47 and 0.00) instead of approximate ~.

**L304** 4.3 Are these results calculated after the slope correction or without slope correction?

**L342** Please provide the NMAD for Morse Creek snow-free terrain.

L363 « At the sites in this study, the individual ICESat-2 » comma.

L377 « In this study, » comma.

L401 « ICESat-2 observations are space »?

L421 « including snow-free terrain » is it a problem ? Snow-free terrain is part of the snow depth variability.

**L435** « and for snow depths > 0.5 m » I am not very convinced by this conclusion. I understand that the uncertainty of ICESat-2 retrievals, in absolute, does not depend on the snow depth value. Thus, an ICESat-2 snow depth of 0.25 m (+-0.5 m) is just as informative to me as 2.00 m (+-0.5 m). It is not because the measurement cannot be distinguished from 0, that it is not informative.

**L446** It results a bit odd to do this analysis over the Idaho state. I doubt that the northern borders follow a relevant natural element. It could be interesting to do this analysis over an area free from subjective human borders (e.g. large area buffered around the AOI or over the mountain ranges to which the AOI belong as defined by the Global Mountain Biodiversity Assessment)

https://www.gmba.unibe.ch/tools/definitions/mountain inventory v1/index eng.html

L456 delete «]»

**L460** What method is used to produce 3DEP? Any source?

**L484** « As previously identified, there is a progressive negative slope bias in the ICESat-2 data (Fig. 4, Deschamps-Berger et al. 2023; Enderlin et al. 2022) » I think the bias is getting progressively positive in Deschamps-Berger et al. (2023). Which does not change the general similarity of this result.

**Figures and Tables**

Table 2. Snow depth R2 = 0.00?

Table 4. I could be nice to have this table plotted for a supplementary figure.

Fig. 1. Make sure that writtings have a sufficient size (lat-lon in western US inset and UTM coordinates on the right pannels). In the legend, write HW21 in full letterrs, maybe even « Avalanche prone... »

Fig. 2. Add grids and make the x-ticks equivalent to the y-ticks.

Not mandatory at all, but it would be nice to add the borders of all the countries encountered in the inset.

Fig. 3. Text is small. Top maps as well. The legend of the histogram might be easier to understand by making the symbol for « Filtered to slopes below 20 degrees » in black (not a colour of the plot) and add an item for « All slopes » (i.e. full line).

Fig. 4a. It is not necessary to modify the plot but cumulative distribution might be easier to interprete as it gives a sense of what proportion of a site is above a given slope.

I would add title on each pannel and horizontal grid lines on b and c.

- Fig. 5. Add grid and make the horizontal 0 m thicker.
- Fig. 6. The text and the plots seem small.

Fig. 7. Add the grid lines on the plot. It is nice that the drop in snow depth at 2000 m in Mores Creek is caught by ICESat-2. Could it be worth mentionning in the text?

---

## Referee Comment (RC2)

**Review**

**Improved workflow for customized ICESat-2 ATL06 elevations captures seasonal mountain snow depths at sub-kilometer scale**

The paper explores how ICESat-2 satellite data can be used to estimate mountain snow depth more accurately by comparing satellite elevations with high-resolution snow-free terrain models. The authors show that with careful processing—such as reducing positioning errors and adjusting for terrain effects—ICESat-2 measurements can closely match ground and airborne observations. The study finds that the satellite performs best in areas with moderate slopes and deeper seasonal snow, and suggests that many mountain regions have conditions suitable for reliable ICESat-2 snow-depth observations. This approach could broaden the use of ICESat-2 for monitoring snowpack and supporting water-resource modeling.

**General comments:**

I find that the paper is well written and that thorough analysis has been performed, identifying limitations in using ICESat-2 for generating snow depth in mountainous areas, while also showing where it can be used. However, I would like to see more descriptions of the general way the processing is done, especially the generation of the hybrid ATL06 product. That description is currently lacking in my view but can easily be fixed. It would also be of interest to include surface classification directly from the number of return photons inside each segment. That would avoid, in my opinion, fully relying on imagery as I understand it, and instead use the inherent physics of the measurements to suppelemt the analysis.

**Line-by-line comments:**

L53: Should this not be 17 m instead of 11 m?

L56: "Comparing ICESat-2 data to an independently collected snow-free DTM introduces additional geolocation errors." Can you state more specifically what you mean and why?

L70: 17 m or 11 m?

L73: "ICESat-2 returns have a geolocation uncertainty of ~4.4 m." Add the error, which is ±6 m, and the fundamental product you are referring to.

L98: I would like some more details of the hybrid data product, as this is important for the study. I think at least a paragraph or two should be dedicated to that purpose to explain how the data is generated.

L176: "The snow-free ICESat-2 height residuals, h_residual, are the difference between ICESat-2 and DTM ground elevations when and where snow was not observed in near-coincident satellite imagery." How were the snow-free conditions determined from the satellite imagery?

L79: ~11 or 17 m?

L181: Why is the "n_fit_photon" not used to calculate when you have snow or snow-free conditions, or used in combination with the imagery? The classification will be quite clear, as the number of return photons can be used to easily separate the two types of returns.

L192: Can you provide some more justification for why "h_mean" is used and not "h_li," for the reader to get a better grasp of why it's important to use it?

L206: Same question as before—can you use the photon count for each segment to determine snow-free conditions?

Figure 2: The text in the figure is very small, so I suggest increasing the font size to make it more visible.

L224: Can you mention the methods that were tested, so the reader does not need to go into the supplement?

L226–L234: Are these co-registrations different from the ones in the appendix?

L276: "Which is more than double the expected precision (4.4 m) of ICESat-2 geolocation." The estimated standard deviation of the error is, however, 6 m, which would still fit within the 1-sigma error. I would not expect you to find an expected precision of 4.4 m, especially in regions of steep terrain.

L284: "We find that ICESat-2 snow depth has a negative bias of ~0.6 m and uncertainty of ~1 m regardless of co-registration approach." So, is there a need to apply the co-registration if these biases still exist?

L305: I would highly suggest that you perform a simple correlation-length analysis of the differences to get an idea of what the optimal comparison radius would be. That would better inform the maximum distance at which you can calculate statistics. Or at least provide a figure of the statistics as a function of your smoothing length (100 m, 500 m, 1000 m, and 5000 m). The optimal smoothing length would most likely be correlated with the average slope magnitude at each site.

L360: Could it also be related to the fact that applying time-variant co-registration reduces the number of samples available and biases the dataset toward specific slope/topographical regions, increasing the noise in the registration? Maybe looking at the number of return photons can help reduce this issue by reducing the impact of mixed surface types where snow and snow-free terrain overlap.

L416: How large are these negative values? To reduce the risk of biasing the snow depth when removing SD < 0, could you allow for smaller negative values to be kept, perhaps within some limit or error?

L430: I think grouping them into elevation zones rather than horizontal distance bins would be a more effective approach, as you will increase data density. That's why I suggested calculating the spatial autocorrelation: you can use that to first get all data within that distance and then group them in elevation bands.

---

## Author Comment (AC1)

**Review of**

**Improved workflow for customized ICESat-2 ATL06 elevations captures seasonal mountain snow depths at sub-kilometer scale**

**Zikan et al. 2025-4813**

**General comment**

This article presents a workflow using ICESat-2 elevations and reference snow-off digital terrain models to map snow depth in four mountainous study sites in the western US over a total of 586 km². The accuracy and precision of the retrieval is evaluated with reference snow depth from automatic weather station and from airborne lidar. The benefits of several steps of the workflow is tested (slope related correction, negative values filtering, coregistration strategy). An empirical definition of areas where snow depth surveyed with ICESat-2 is likely to be informative and valuable is proposed and used to estimate the coresponding area in the Idaho state (US).

The article reads well. The methodology and results are well presented. Some results are insightfull, such as the impact of the coregistration strategy and of the slope related correction. However, the quantification of the improvement brought by this specific workflow in comparison with previous studies is lacking. This work builds upon workflows presented in previous work which it aims to improve (Enderlin et al., 2022 ; Besso et al., 2024). I would expect a part of the discussion to be dedicated to a comparison of uncertainty and precision metrics to highlight the improvement brought by this workflow. Without this, it is hard to evaluate the benefits of it. A comparison with the results in Deschamps-Berger et al. (2023) and Chen et al. (2025) could also be included since they share a common general framework of differencing with external elevation models. Regarding the comparison with Besso et al. (2024), see the comment about Line 103-105.

César Deschamps-Berger

César,

Thank you for your constructive and insightful review, we greatly appreciate the opportunity to improve this work based on your expertise. We plan to add more detail on previous studies and an additional section to the discussion dedicated to comparison to previous work to expand our comparison with previous ICESat-2 studies. Please see the proposed addition to the discussion below and our detailed

responses to your individual comments for the specific changes we propose to make.

Karina Zikan

Proposed addition to the discussion:

"The most pertinent comparisons to previous studies are to Enderlin et al. (2022), which studied Reynolds Creek, and Besso et al. (2024), which used ATL06_SR in a mountain environment. We see a marked improvement in the Reynolds Creek ICESat-2 NMAD from 0.95 m to 0.41 m that we attribute largely to the smaller segment size of ATL06_SR relative to ATL08 as used in Enderlin et al (2022). Through comparison to AWS snow depths in a different mountain environment, Besso et al. (2024) found median ICESat-2 snow depth RMSE of 0.28-0.62 m and we obtain a comparable RMSE range of 0.21-0.43 m, excluding Dry Creek. We also calculate lower or comparable NMAD values at Reynolds Creek, Banner Summit, and Mores Creek to the NMAD for snow depths in Deschamps-Berger et al. (2023). The small decrease in NMAD is likely due to the inclusion of ATL08 vegetation filtering in ATL06_SR, in line with the conclusions of Besso et al. (2024). However, we find our snow depth NMAD estimates still exceed those obtained over the flatter (generally < 10°) and open terrain at Creamer's Field and Toolik in Fair et al. (2025), supporting the strong dependence of ICESat-2 snow depth accuracy on slope even with optimized processing."

**Minor comments and suggestions**

Suggested modifications are in ***bold italic***.

**L14** Provide the RMSE and R2 with 2 decimals precision, like in Table 4.

We will provide the RMSE and R2 with 2 decimal precision as recommended.

**L25** Order chronologically the citations.

We will go through the manuscript and make sure all citations are in the cryosphere style.

**L28** Precise whom Decadal Survey it is or turn the sentence in passive mode « Goals of global snow water equivalent (SWE) at 1–4 km resolution and ~100 m SWE resolution ***were set*** for mountain regions to meet snow observation needs for water management (Decadal Survey...) »

Good point, the 2017-2027 Decadal survey is specifically a guidance for US science agencies. Since ICESat-2 is a NASA satellite we would like to highlight

how it can be used to approach goals set for the agency so we will rewrite L28 to precisely state the providence of the decadal survey.

Current text: "Given the high spatial variability of snow, the 2017 Decadal survey calls for 1–4 km global snow water equivalent (SWE) resolution and ~100 m SWE resolution in the mountains to meet snow observation needs for water management (National Academies of Sciences, Engineering, and Medicine (U.S.), 2018)"

Suggested rewrite: "Given the high spatial variability of snow, the United States 2017-2027 Decadal Survey for Earth Science and Applications from Space calls for 1–4 km global snow water equivalent (SWE) resolution and ~100 m SWE resolution in mountain regions to meet snow observation needs for water management (National Academies of Sciences, Engineering, and Medicine (U.S.), 2018)"

**L31** « ; » replace with « but » or another similar conjunction.

We will replace ; with a conjunction. Please see the next comment for our full suggested rewrite of L31.

**L31** Maybe also state that SWE cannot be directly observed at the relevant scale and resolution?

We agree it would be good to specifically state that SWE cannot currently be observed at the relevant scale and resolution. We propose to rewrite L31 as follows

Current text: "SWE is calculated from snow depth and density; currently neither snow depth nor snow density observations are available at the scale and resolutions called for in the 2017 Decadal survey."

Suggested rewrite: "Although SWE can be calculated from snow depth and density, neither SWE, snow depth, nor snow density observations are available at the called for scale and resolutions."

**L34** « In this paper**,** » comma.

We will add a comma as suggested.

**L35** It is implied but state clearly that this refers to airborne and terrestrial lidar.

We will replace "Lidar" with "Airborne and terrestrial lidar" as suggested.

**L37-38** no need to repeat « airborne and terrestrial Lidar » the second time?

Good point, we will remove the repetition.

**L41** Some quantified values of the ICESAt-2 snow depth error found in previous studies would be welcome. See for instance, Fair et al. (2025).

We propose to add the following table summarizing previous studies.

Table draft:

| | ICESat-2 product | Study site | Reference DTM | Uncertainty metric | Uncertainty | | | Snow depth validation data sets | |
|---|---|---|---|---|---|---|---|---|---|
| Enderlin et al. | ATL08 | Reynolds Creek, ID | Areal lidar (1m) | MAD | 0.64 m | < 5° slope: ~0.2 m | > 20°slope: >1 m | In situ weather stations | |
| Dechamps-berger et al. | ATL06 | Upper Tuolumne, CA | ASO Areal lidar (15m) | NMAD | 0.89 m | < 10° slope: 0.5m | | ASO snow depth maps (15m) | |
| | | | Pleiades statellite photogrametry (15m) | | 1.01 - 1.16 m | | | | |
| | | | Copurnicus DEM (30m) | | 3.00 m | | | | |
| Besso et al. | ATL06_SR | Tuolumne, CA | ASO Areal lidar (3m) | Median snow depth RMSE | 0.18 m | | | In situ weather stations | ASO snow depth map (3m) |
| | | Methow Valley, WA | Areal lidar (1m) | | 0.33 m | | | In situ weather stations | |
| Chen et al. | ATL03 | Tuolumne, CA | ASO Areal lidar (3m) | Mean adsolute error (MAE) | 0.31 m | < 10° slope: 0.17 m | | In situ weather stations | ASO snow depth map (3m) |
| Fair et al. | ATL06, ATL08, ATL06_SR | Creamer's Field, AK | UAF Areal lidar (0.5 m) | NMAD | 0.20 m, 0.16 m, 0.12 m | | | UAF lidar snow depth map (0.5 m) | |
| | | Toolik, AK | | | 0.11 m, 0.12 m, 0.14 m | | | | |

**L43** « for static terrain ...for spatio-temporally evolving terrain » This formulation got me confused and is innacurate since Lu et al. (2022) also apply their method on land masses.

Thank you for the correction, we will remove the inaccurate "static terrain" and "spatio-temporally evolving terrain" language.

**L64** « can be applied in shallow to moderately sloped terrain » I thought that one of the conclusion of the article is that snow depth below 0.5 m are not well monitored with ICESat-2?

Apologies for the confusion, we are only discussing slope ranges for which ICESat-2 is usable in L64. For clarity we will change "shallow to moderately sloped terrain" to "shallow- to moderately- sloped terrain".

**L65** « covered by the USGS 3D Elevation Program (3DEP) or the Swiss national DTM (swissALTI3D) » Move to Discussion.

We will move this to L460 in the discussion. Please see our response to the L460 comment for our proposed edits.

**L70** Say that the high-frequency acquisition provides continuous acquisitions along-track.

Current text: "ICESat-2 measures elevations along three pairs of weak and strong beam tracks"

Suggested rewrite: "The high-frequency acquisition of ATLAS provides continuous along-track photon returns along three pairs of weak and strong beam tracks."

**L72** « Launched in 2018, ICESat-2 has a 91 day polar orbiting cycle but it points off-nadir outside of the polar regions **to increase coverage and to map vegetation**, repeating reference ground tracks in mid-latitudes every 3 years **only**. » Suggestion.

This is good context, we will rewrite L72 as suggested.

**L74** « of repeat reference ground tracks » => « tracks »

We will change L74 as suggested.

**L75** I would turn this sentence in passive mode : « the spatio-temporal restriction **can be circumvented** ». And cite the works which did that.

We will turn this sentence into passive tense and add citations as suggested.

**L79** I would split this sentence in two. If I understood correctly, something along : « We conduct our analysis using observations from four Idaho study sites for which **ICESat-2** acquisitions were tasked **to cover/overpass** an automated weather station within each site. **This resulted in shifting the satellites tracks within a 30 km radius around the stations and enabled the comparison of satellite and** in situ snow depth data from 2020–present, as described below »

Thank you for the writing suggestion, we agree this makes L79 clearer. We will rewrite L79 similar to how you suggest.

Current text: "We conduct our analysis using observations from four Idaho study sites for which acquisitions were tasked for every track within 30 km on an automated weather station with in situ snow depth data from 2020–present, as described below."

Proposed rewrite: "We conduct our analysis using observations from four Idaho study sites for which ICESat-2 acquisitions were tasked to overpass an automatic weather station (AWS) within each site. ICESat-2 was tasked so that every track within a 30 km radius of the AWSs were shifted to target the AWS, thus enabling the direct comparison of ICESat-2 and situ snow depth data from 2020–2024, as described in section 3."

**L81** « 2020–**present** » find a formulation that will remain true in a few years

As included in the proposed rewrite above, we will change <<2020-present>> to <<2020-2024>> so the date range will remain accurate in the future.

**L84** « ATL06 » as I understand the workflow, ATL06 is not used but an ATL06-like product (L100). Should it say here that this study uses a hybridized product based on ATL08 and ATL03 resulting in a ALT06-like product ?

We agree that the current text is confusing so we will rewrite the sentence as follows for clarity.

Current text: "For this study we use a hybridized data product based on two higher level ICESat-2 products: the Land Water Vegetation Elevation product (ATL08) (Neuenschwander et al., 2023) and the Land Ice Elevation product (ATL06) (Smith et al., 2023)."

Proposed rewrite: "For this study we use a hybridized ATL06-like data product calculated from the Global Geolocated Photon Data product (ATL03) and based on two higher level ICESat-2 product algorithms: the Land Water Vegetation Elevation product (ATL08) photon classification (Neuenschwander et al., 2023) and the Land Ice Elevation product (ATL06) surface elevation algorithm (Smith et al., 2023)."

**L89** NMAD. You may want to cite the original article as well, Höhle and Höhle (2009).

Höhle, J., & Höhle, M. (2009). Accuracy assessment of digital elevation models by means of robust statistical methods. *ISPRS Journal of Photogrammetry and Remote Sensing*, *64*(4), 398-406.

Thank you for catching the omission, our original intent was to cite Höhle and Höhle (2009) as well so we will add the citation.

**L91** « where there is no vegetation but ***where*** differences in surface elevation must be resolved on the order of centimeters in order to capture important variations in snow and ice volume »

We propose to split the lack of vegetation and the need for high vertical resolution into two separate sentences so these two important characteristics of ATL06 can be emphasized more clearly.

Suggested rewrite of L91: "ATL06 is designed for observing ice sheets and glaciers where differences in surface elevation must be resolved on the order of centimeters in order to capture important variations in snow and ice volume. As these land-ice surfaces are vegetation free, ATL06 does not include vegetation filtering."

**L95** « vertical accuracy is expected to exceed » not sure if it exceeds in absolute value ( Accuracy_ATL06 > Accuracy_ATL08) or in quality (ATL06 better than ATL08, Accuracy_ATL06 < Accuracy_ATL08)

You're right, this is confusing as written. The intention was that it exceeds in quality, we will rewrite the sentence to remove this ambiguity.

Current text: "In rough snow-covered terrain, comparable to our mountain study sites, ATL06's vertical accuracy is expected to exceed the aforementioned ATL08 uncertainty."

Suggested rewrite: "In rough snow-covered terrain, comparable to our mountain study sites, ATL06 is expected to have a better vertical accuracy than the aforementioned ATL08 product.

**L100** Move (Besso et al., 2024) and (Shean et al., 2025) together at the end of the sentence.

We will move the citations as recommended.

**L101** I would avoid mentioning the « SlideRule atl06 function » which is a bit cryptic. Simply list what corrections and filters are applied or not.

We plan to largely rewrite this section in response to both this suggestion and comments from the other reviewer. Please see our proposed rewrite of L98-102 and section L189-192 below,

Proposed L98 changes: "In this study we make use of the strengths of both algorithms using a hybridized data product (ATL06_SR) (Besso et al., 2024, Fair et al., 2025) that incorporates ATL08 vegetation filtering and the ATL06 algorithm into an ATL06-like product. ATL06_SR is calculated by applying the ATL06 function to ATL08-identified ATL03 ground photon returns instead of the ATL03-identified ground photon returns. The generation of ATL06_SR is discussed in more detail in section 3.1. As applied in this paper, ATL06_SR includes ATL08's vegetation filtering but does not include the first photon bias correction, which can result in up to ~2 cm of bias, or the transit pulse shape bias, which can result in up to ~1 cm of bias."

Proposed 3.1 (L189-192) changes: "We use the ATL06_SR product (Besso et al., 2024) for all available ICESat-2 data acquired from October 2018 to April 2024 within the boundaries of the four study sites. To calculate ATL06_SR, we applied the ATL06 function to ATL08 ground-classified ATL03 photons (as in Besso et al., 2024). We calculated ATL06_SR using the SlideRule Earth data processing package which allows for rapid, cloud-based processing of the ATL03 photon cloud with customized control of the ATL06 algorithm parameters (Shean et al., 2025). For this study ATL06_SR was calculated using ATL08 ground-classified ATL03 photons using otherwise default ATL06 parameters: a 40 m segment length, a step size of 20 m, a minimum along-track spread of 20 m, a maximum of 6 iterations, and a minimum of 10 ATL08 ground classified photons. The resulting ATL06_SR product therefore has an elevation estimate every 20 m."

**L103-105** « ATL06_SR derived snow depths have a Root Mean Square Error (RMSE) of 0.18 m in Tuolumne Basin compared to a 3 m resolution ASO DTM and a RMSE of 0.33 m in the Methow Valley in Washington compared to a 1 m resolution airborne lidar DTM (Besso et al., 2024). » These value of RMSE are misleading as they are not the RMSE at the ATL06_SR resolution of 20 m. This is the RMSE of the ensemble of median differences calculated for each day with ICESat-2 data. This can be verified in the code from Besso et al. (2024). For instance for the Methow Valley :

https://github.com/bessoh2/icesat2_sr/blob/main/methow_valley/notebooks/comparison_to_snotel.ipynb

In the box 56, the RMSE (0.33 m) is calculated on the object *comp_df*, itself defined in box 54 and 55. It is filled with median differences, one for each acquisition date.

When pushing the comparison of these results with Besso et al. (2024), this code might be useful to ensure the consistency of the variables compared.

Thank you for calling our attention to the details regarding these uncertainty estimates. We propose to clarify the language in the text so that it is clear that these uncertainty metrics are not calculated using all the ATL06_SR segments, but are instead the RMSE of the daily median differences. We will also add a comparison of median ICESat-2 snow depth RMSE between this study and Besso et al. 2024 to the discussion (Please see the proposed addition to the discussion at the beginning of this response document).

**L105** « Thus**,** » comma.

We will add a comma as recommended.

**L111-114** That is a lot of acronyms, institutions name, and programs name. Could it be rather put in a table (acquisition date, resolution, program, provider...)? Possibly in supplement.

« Each site has a high-resolution snow-free airborne lidar DTM  *freely*(?) available *acquired during various campaigns (Table XX)*. »

These information are repeated in the study sites description. Maybe that is even enough then.

We agree the current text is overwhelming. To fix this we will just list the DTM sources in the site descriptions. We will also edit L111 as recommended but with a pointer to the following site descriptions rather than a table.

**L119** « 1.3 cm » source ?

We will add citations for the ultrasonic snow depth sensors used on SNOTEL AWSs, the Judd Communications depth sensor and the Sommer USH-9. Additionally, for consistency with the rest of the paper we will report the precision in meters instead of centimeters.

**L122** 2.2.1-4 Could the basins presentation be ordered from north to south ? or another geographical logic ?

We propose to order the study sites by size. The new order of sites would be Reynolds Creek, Banner Summit, Dry Creek, and Mores Creek. We believe this order makes sense given the flow of the paper; Reynolds Creek and Banner Summit have many similar attributes, while Dry Creek deviates in surface roughness and typical snow depths but is comparable in size to Banner. Mores Creek is by far the smallest site and is the only site with airborne lidar data so it feels appropriate to highlight Mores Creek last.

As part of this change we will reorder sections 2.2.1-2.2.4, reorder any instance where the sites are listed in order, and reorder the sites in each figure to match the new order.

**L127** « ~1300-1600 m to 1669–2013 m » keep the same precision for both ranges.

We will report the date ranges as "~1300-1600 m to ~1700-2000 m" as recommended.

**L186** « ICESat-2 snow depth precision or accuracy » I see the point in distinguishing precision and accuracy but please provide the definition used in this article. Is one the bias, the dispersion, the combination of both… ?

By precision we mean the uncertainty and by accuracy we mean the bias.

Suggested rewrite: "ICESat-2 snow depth precision (uncertainty) or accuracy (bias)"

**L191** « a *minimal number of* photon threshold » ?

Yes this is the minimum photon return threshold, we will rewrite L191 to specify.

**L236** « The reference elevation for each ATL06_SR point is the mean elevation of the DTM within the corresponding ATL06_SR segment area (a 11 m by 40 m rectangle, oriented along the ICESat-2 track) » nice effort.

Thank you, we will leave L236 as it is.

**L238-240** « Although the terrain metrics... » I do not understand this. Isn't it the case for ATL06 products ?

Yes, that's true. We thought it was important to state that the metrics calculated from the DTMs are assigned to segment centroid coordinates in the same fashion as the ATL06_SR product. To clarify this point we will slightly rewrite L238-240.

Current text: "Although the terrain metrics are extracted from within the entire ATL06_SR segment area, the terrain metrics are nominally assigned the same segment centroid coordinates as the ATL06_SR elevation to facilitate analysis."

Proposed rewrite: "Although the DTM derived terrain metrics are extracted from within the entire ATL06_SR segment area, the DTM terrain metrics are nominally assigned the same segment centroid coordinates as the ATL06_SR elevation to facilitate analysis."

**L251** « To evaluate the scale... » I am unsure how the smoothing is done on ICESat-2 elevation : alongtrack or within a square, circle shape? The latter would make the number of data points smoothed variable.

The smoothing is done by taking the median snow depth within a certain radius of the AWS, it is the radius of the circular area that is referred to as the smoothing length. To clarify this definition, we propose the following rewrite of L251,

Current text: "To evaluate the scale at which ICESat-2 best captures snow depth patterns we calculate the median ICESat-2 snow depth within various lengths, herein we refer to this length as the smoothing length"

Proposed rewrite: "To evaluate the scale at which ICESat-2 best captures snow-depth patterns we calculate the median ICESat-2 snow depth within a circle of various radius lengths centered on the AWS, herein we refer to the radius of this circle as the smoothing length"

**L271** « and precision then present » add a comma somewhere ?

We will add the missing comma between "precision" and "then".

**L282** « *T*able 2 »

We will correct the capitalization error.

**L287** The exact R2 values can be provided (0.47 and 0.00) instead of approximate ~.

We will report exact values as recommended.

**L304** 4.3 Are these results calculated after the slope correction or without slope correction ?

All of these results are calculated after the slope correction is applied. We will add a sentence to state that all results in 4.3 are calculated after the slope correction is applied.

**L342** Please provide the NMAD for Morse Creek snow-free terrain.

Good point, we will add the NMAD for Morse Creek snow-free terrain (1.08 m) and a pointer to table 2 where these statistics are reported.

**L363** « At the sites in this study**,** the individual ICESat-2 » comma.

We will add a comma as recommended.

**L377** « In this study**,** » comma.

We will add a comma as recommended.

**L401** « ICESat-2 observations are space » ?

Thank you for catching the typo, we will remove it. L401 was intended to read "At Dry Creek, where the majority of slopes are > 20°, ICESat-2 snow depths are unreliable."

**L421** « including snow-free terrain » is it a problem ? Snow-free terrain is part of the snow depth variability.

That is an excellent point, we will remove "including snow-free terrain".

**L435** « and for snow depths > 0.5 m » I am not very convinced by this conclusion. I understand that the uncertainty of ICESat-2 retrievals, in absolute, does not depend on the snow depth value. Thus, an ICESat-2 snow depth of 0.25 m (+-0.5 m) is just as informative to me as 2.00 m (+-0.5 m). It is not because the measurement cannot be distinguished from 0, that it is not informative.

I see we did not explain our argument well, hopefully we can address your concerns on this point. We agree that a snow depth of 0.25 m (+-0.5 m) is just as informative as a snow depth of 2.00 m (+-0.5 m). We believe that ICESat-2 snow depths are less reliable over shallow snowpacks due to the impact of low, shrub vegetation which is common in the sites used in this study. This low vegetation is depressed and smoothed in deeper snowpacks. We propose to remove the conclusion about snow depth from L435 and add a new section into L433-445 to discuss the snow depth and vegetation impact separately. Please see the proposed rewrite below with the new section in bold.

Proposed rewrite of L433-445: "ICESat-2 snow depth mapping is promising, capturing spatial and temporal distribution patterns at scales of 100–1000 m; however, ICESat-2 snow depths are not reliable in all conditions. Based on the results for our study sites, we recommend using ICESat-2 snow depth mapping in regions with a majority of slopes < 20° *<<removed text: and for snow depths > 0.5 m>>*. **For instance,** at Mores Creek (Fig. 7a), when ICESat-2 returns from steep slopes (> 20°) are filtered, snow depth uncertainty is ~0.3 - 0.7 m. Although ICESat-2 snow depth uncertainties can be as low as 0.2 m where the majority of slopes are < 10°, restricting ICESat-2 application to such shallow-sloped regions will limit their usefulness. For steeper-sloped terrain, the slope-related bias can be minimized and uncertainty can be reduced by up to 30% if a site-specific slope correction is calculated for and applied to ATL06_SR, so ICESat-2 can be used for snow depth observation in more complex terrain with appropriate spatial averaging. **As with all lidar techniques, care must be taken when interpreting snow-covered ATL06_SR elevations with respect to a reference DTM in regions where low, matted vegetation is present (Tinkham et al. 2015, May et al., 2025). Deep snow covers and smooths this low vegetation and small-scale topographic roughness resulting in a clearer snow signal relative to regions with a shallow snowpack (Deems et al. 2013, Filhol et al. 2019) but uncertainties are still greater than for bare or forested terrain. For example, though Mores Creek and Banner Summit have similar slope distribution the snow depth uncertainties for Mores Creek are slightly larger than for Banner Summit likely due to the abundance of low bushes in close proximity to the More Creek AWS. Despite similar vegetation near the Dry Creek AWS, the snow depth uncertainty for Mores Creek is ~1 m less than for Dry Creek, due to the combined effects of more compressed vegetation and shallower slopes at Mores Creek on ATL06_SR uncertainties.** Even with slope corrections, sites like Dry Creek where steep slopes (majority > 20°) and bushy terrain increase snow depth uncertainty to 1.5 m and snow depth error to 1.2 m are ill-suited for ICESat-2 observation."

**L446** It results a bit odd to do this analysis over the Idaho state. I doubt that the northern borders follow a relevant natural element. It could be interesting to do this analysis over an area free from subjective human borders (e.g. large area buffered around the AOI or over the mountain ranges to which the AOI belong as defined by the Global Mountain Biodiversity Assessment)

https://www.gmba.unibe.ch/tools/definitions/mountain_inventory_v1/index_eng.html

We agree this would be an improvement; we propose to make this plot using the USGS Pacific Northwest region watershed boundary instead of the Idaho state border.

Draft of new slope uncertainty plot with a proposed map extent inset:

[Figure]

**L456** delete « ] »

Thank you for catching the typo, we will remove it.

**L460** What method is used to produce 3DEP ? Any source ?

Proposed rewrite of L460 incorporating the moved section of L65.

Proposed rewrite: " For example, the USGS 3D Elevation Program (3DEP) has publicly available 1 m resolution lidar DTM coverage for over 70% of the Western

United States and is working toward 100% coverage. Other mountain regions also have high resolution lidar DTM coverage, such as the Swiss national DTM (swissALTI3D)."

**L484** « As previously identified, there is a progressive negative slope bias in the ICESat-2 data (Fig. 4, Deschamps-Berger et al. 2023; Enderlin et al. 2022) » I think the bias is getting progressively positive in Deschamps-Berger et al. (2023). Which does not change the general similarity of this result.

Thank you for the correction, we will fix the mistake.

Proposed rewrite: "As previously identified at Reynolds Creek, there is a progressive negative slope bias in the ICESat-2 data (Fig. 4, Enderlin et al. 2022). Deschamps-Berger et al. 2023 also found a progressive, though positive, slope bias."

**Figures and Tables**

Table 2. Snow depth R2 = 0.00 ?

This is correct. Somewhat shockingly there is no relationship between the ICESat-2 and airborne snow depths when using an individual co-registration.

Table 4. I could be nice to have this table plotted for a supplementary figure.

In response to this and a recommendation from the second reviewer we are planning to move table 4 to the supplement and replace it with this figure plotting the snow depth R2 and RMSE by smoothing length.

Draft of new figure:

[Figure]

Fig. 1. Make sure that writtings have a sufficient size (lat-lon in western US inset and UTM coordinates on the right pannels). In the legend, write HW21 in full letterrs, maybe even « Avalanche prone... »

We will increase text size and update the legend as recommended

Fig. 2. Add grids and make the x-ticks equivalent to the y-ticks.

Not mandatory at all, but it would be nice to add the borders of all the countries encountered in the inset.

We will make the x- and y-ticks equivalent and add a grid.

Assuming the second suggestion is regarding figure 1, we will also add the borders of Canada and Mexico to the map inset in figure 1.

Fig. 3. Text is small. Top maps as well. The legend of the histogram might be easier to understand by making the symbol for « Filtered to slopes below 20 degrees » in black (not a colour of the plot) and add an item for « All slopes » (i.e. full line).

We will make the plot bigger, making the text more legible, and update the legend as recommended.

Fig. 4a. It is not necessary to modify the plot but cumulative distribution might be easier to interprete as it gives a sense of what proportion of a site is above a given slope.

We propose to add this cumulative distribution figure as a supplement.

New figure draft:

[Figure]

I would add title on each pannel and horizontal grid lines on b and c.

We will update the figure as recommended.

Fig. 5. Add grid and make the horizontal 0 m thicker.

We will update the figure as recommended.

Fig. 6. The text and the plots seem small.

We will make the text bigger

Fig. 7. Add the grid lines on the plot. It is nice that the drop in snow depth at 2000 m in Mores Creek is caught by ICESat-2. Could it be worth mentionning in the text ?

Proposed addition to L386-388 (Bold): "Our comparison with these data indicate that ICESat-2 systematically underestimates observed snow depth but accurately captures orographic distribution patterns across the 38 km2 area, **including the drop in snow depth at 2000 m elevation** (Table 2, Fig. 3, Fig. 7).

---

## Author Comment (AC2)

**Review**

**Improved workflow for customized ICESat-2 ATL06 elevations captures seasonal mountain snow depths at sub-kilometer scale**

The paper explores how ICESat-2 satellite data can be used to estimate mountain snow depth more accurately by comparing satellite elevations with high-resolution snow-free terrain models. The authors show that with careful processing—such as reducing positioning errors and adjusting for terrain effects—ICESat-2 measurements can closely match ground and airborne observations. The study finds that the satellite performs best in areas with moderate slopes and deeper seasonal snow, and suggests that many mountain regions have conditions suitable for reliable ICESat-2 snow-depth observations. This approach could broaden the use of ICESat-2 for monitoring snowpack and supporting water-resource modeling.

**General comments:**

I find that the paper is well written and that thorough analysis has been performed, identifying limitations in using ICESat-2 for generating snow depth in mountainous areas, while also showing where it can be used. However, I would like to see more descriptions of the general way the processing is done, especially the generation of the hybrid ATL06 product. That description is currently lacking in my view but can easily be fixed. It would also be of interest to include surface classification directly from the number of return photons inside each segment. That would avoid, in my opinion, fully relying on imagery as I understand it, and instead use the inherent physics of the measurements to suppelemt the analysis.

We thank the reviewer for their detailed and constructive comments, we are glad to improve the work herein. We plan to revise and expand our description of the hybrid ATL06_SR product and to include a direct pointer to Besso et al. 2024 and Shean et al. 2025 where the product is described in greater detail, please see our response to L98 for more detail. We additionally performed new smoothing length and auto-correlation analysis to give more depth to our smoothing length discussion. We also propose to add additional detail on the return photons to better explain our decision to use the NSDI imagery as opposed to n_fit_photon as we initially attempted. Please see our detailed responses to the individual comments for the specific changes we propose to make.

Karina Zikan

**Line-by-line comments:**

L53: Should this not be 17 m instead of 11 m?

We chose to use the ICESat-2 mean effective laser footprint diameter of 10.9 m ± 1.2 m rounded to 11 m found by Magruder et al. 2021 instead of the 17 m diameter estimate. For clarity, we will remove reference to the footprint diameter from L53 and expand L70 to explain the use of the 11 m mean effective laser footprint diameter. Please see the L70 comment for the specific proposed rewrite.

L56: "Comparing ICESat-2 data to an independently collected snow-free DTM introduces additional geolocation errors." Can you state more specifically what you mean and why?

To add more explanation for the geolocation errors we will rewrite L56 as follows,

Current L56 text: "Additionally, comparing ICESat-2 data to an independently collected snow-free DTM introduces additional geolocation errors (Enderlin et al., 2022; Hugonnet et al., 2022; Nuth and Kääb, 2011)."

Proposed rewrite: "Additionally, if the ICESat-2 data and independently collected snow-free DTM are not properly aligned geospatially the resulting geolocation offset between the two datasets will introduce geolocation errors. The magnitude of these geolocation errors depends on the slope and aspect of the underlying terrain relative to the direction of the geolocation offset (Enderlin et al., 2022; Hugonnet et al., 2022; Nuth and Kääb, 2011)."

L70: 17 m or 11 m?

To clarify the use of the 11 m mean effective laser footprint diameter we will rewrite L70 as follows,

Current L70 text: "Each pair of beams has a beam footprint of ~11m (Magruder et al., 2021), an intra-pair separation of 90 m, and an inter-pair separation of 3.3 km (Neumann et al., 2019)"

Proposed rewrite: "Each pair of beams has an intra-pair separation of 90 m, and an inter-pair separation of 3.3 km (Neumann et al., 2019). After launch, the mean effective laser footprint diameter of ATLAS was found to be ~11 m (Magruder et al., 2021)"

L73: "ICESat-2 returns have a geolocation uncertainty of ~4.4 m." Add the error, which is ±6 m, and the fundamental product you are referring to.

We will add the error and the ATL03 product we are referring to.

L98: I would like some more details of the hybrid data product, as this is important for the study. I think at least a paragraph or two should be dedicated to that purpose to explain how the data is generated.

We discuss the generation of ATL06_SR in more detail in section 3.1. We will add a pointer to section 3.1 to L98 and provide more detail on the correction and filters used to the ATL06_SR paragraph in 3.1.

Proposed L98 changes: "In this study we make use of the strengths of both algorithms using a hybridized data product (ATL06_SR) (Besso et al., 2024, Fair et al., 2025) that incorporates ATL08 vegetation filtering and the ATL06 algorithm into an ATL06-like product. ATL06_SR is calculated by applying the ATL06 function to ATL08-identified ATL03 ground photon returns instead of the ATL03-identified ground photon returns. The generation of ATL06_SR is discussed in more detail in section 3.1. As applied in this paper, ATL06_SR includes ATL08's vegetation filtering but does not include the first photon bias correction, which can result in up to ~2 cm of bias, or the transit pulse shape bias, which can result in up to ~1 cm of bias."

Proposed 3.1 (L189-192) changes: "We use the ATL06_SR product for all available ICESat-2 data acquired from October 2018 to April 2024 within the boundaries of the four study sites. We refer the reader to Besso et al., 2024 and Shean et al. 2025 for a detailed description of the ATL06_SR product used herein. Briefly, to calculate ATL06_SR, we applied the ATL06 function to ATL08 ground-classified ATL03 photons (as in Besso et al., 2024). We calculated ATL06_SR using the SlideRule Earth data processing package which allows for rapid, cloud-based processing of the ATL03 photon cloud with customized control of the ATL06 algorithm parameters (Shean et al., 2025). For this study ATL06_SR was calculated using ATL08 ground-classified ATL03 photons using otherwise default ATL06 parameters: a 40 m segment length, a step size of 20 m, a minimum along-track spread of 20 m, a maximum of 6 iterations, and a minimum of 10 ATL08 ground classified photons. The resulting ATL06_SR product therefore has an elevation estimate every 20 m."

L176: "The snow-free ICESat-2 height residuals, h_residual, are the difference between ICESat-2 and DTM ground elevations when and where snow was not observed in nearcoincident satellite imagery." How were the snow-free conditions determined from the satellite imagery?

We will edit L176 to mention the NDSI filtering. More detail on the NDSI filtering is in L204-214.

Current text: "The snow-free ICESat-2 height residuals, h_residual, are the difference between ICESat-2 and DTM ground elevations when and where snow was not observed in near-coincident satellite imagery."

Suggested rewrite: "The snow-free ICESat-2 height residuals, h_residual, are the difference between ICESat-2 and DTM ground elevations in snow-free conditions as determined by the Normalized Difference Snow Index (NDSI) calculated from in near-coincident satellite imagery."

L79: ~11 or 17 m?

~11 m

L181: Why is the "n_fit_photon" not used to calculate when you have snow or snow-free conditions, or used in combination with the imagery? The classification

will be quite clear, as the number of return photons can be used to easily separate the two types of returns.

We initially tried to use the n_fit_photon to identify snow cover, however since the number of photon returns is greatly reduced by both vegetation cover and increased slope we found the n_fit_photon was not reliable for separating snow covered and snow free terrain. We chose to use the near-coincident NDSI maps as an independent snow mask unaffected by the terrain metrics we wanted to investigate.

We propose to add the following text after L182 to expand on the choice to use the NDSI maps rather than the n_fit_photon.

"Due to the effect of slope and vegetation cover, which both reduce the number of ground return photons (n_fit_photon), snow-free and snow-covered conditions could not be clearly distinguished using n_fit_photon at the sites studied. A difference in n_fit_photon distribution was observed when slopes were < 10° however we chose to use the independent near coincident NDSI maps to identify snow covered terrain so as not to impact the terrain related controls."

As you can see from these histograms of photon returns in snow-free, summer, and snow-covered conditions while there is a difference in the distribution of photon returns, there is too much overlap between surface cover to cleanly differentiate

between surface conditions.

[Figure]

L192: Can you provide some more justification for why "h_mean" is used and not "h_li," for the reader to get a better grasp of why it's important to use it?

The choice of h_mean was a practical one as Sliderule only calculates h_mean, not h_li. We will add more detail to clarify the use of h_mean.

L206: Same question as before—can you use the photon count for each segment to determine snow-free conditions?

Please see our detailed response above.

Figure 2: The text in the figure is very small, so I suggest increasing the font size to make it more visible.

We will increase figure font sizes.

L224: Can you mention the methods that were tested, so the reader does not need to go into the supplement?

We will add the following text to list the other co-registration algorithms tested,

Proposed rewrite of L224: "In addition to the iterative grid search two other co-registration algorithms were tested and rejected during method development, the Nuth and Kääb (2011) co-registration approach (as used in Deschamps-Berger et al. 2023), and a gradient descent (as used in Enderlin et al. 2022). These are discussed further in Appendix A."

L226–L234: Are these co-registrations different from the ones in the appendix?

L226-234 and the appendix are discussing different aspects of the co-registration. L226-234 is focused on how the data is input into the co-registration algorithm (either aggregated or as individual tracks) while the appendix is focused on other co-registration algorithms we tested in addition to the iterative grid approach (L214-224). Currently we are referring to both parts of the coregistration as the "coregistration approach" which is confusing, to fix this we will refer to the coregistration algorithm as the "co-registration algorithm" and the data input into the coregistration algorithm as the "co-registration approach".

L276: "Which is more than double the expected precision (4.4 m) of ICESat-2 geolocation." The estimated standard deviation of the error is, however, 6 m, which would still fit within the 1-sigma error. I would not expect you to find an expected precision of 4.4 m, especially in regions of steep terrain.

Yes that's a good point, we will remove this statement. On reflection, since our goal of this line is to highlight that the individual coregistration finds larger offsets then the aggregated coregistration we will report the variability in the offset (interquartile range of 3.9 m) rather than the maximum shift offset.

L284: "We find that ICESat-2 snow depth has a negative bias of ~0.6 m and uncertainty of ~1 m regardless of co-registration approach." So, is there a need to apply the coregistration if these biases still exist?

We agree that there is a reasonable argument to be made that, at least with the co-registration methods we tested, co-registration does not improve ICESat-2 results, or not to an extent that it is worth the time and computation effort. We

believe it is still important to report and discuss these results to inform future research and hopefully save future researchers some time. We will add the following text to the discussion in section 5.1 after L353,

"Co-registration remains an unsolved problem. Regardless of co-registration approach, ICESat-2 snow depth maintained a negative bias of ~0.6 m and uncertainty of ~1 m. There is likely a limit to the improvement possible from horizontal co-registration. While horizontal co-registration should not be inherently dismissed because there can be systematic offsets between ICESat-2 and the reference DTM depending on the georeferencing of the reference DTM, the time and computation effort required to perform the horizontal co-registration should be weighed against potential improvements."

L305: I would highly suggest that you perform a simple correlation-length analysis of the differences to get an idea of what the optimal comparison radius would be. That would better inform the maximum distance at which you can calculate statistics. Or at least provide a figure of the statistics as a function of your smoothing length (100 m, 500 m, 1000 m, and 5000 m). The optimal smoothing length would most likely be correlated with the average slope magnitude at each site.

We propose to replace table 4 with a plot of RMSE and R2 by smoothing length and movie table 4 to the supplement. We will update the text regarding smoothing length at each site based on this new figure.

Additionally we plan to add a figure plotting ICESat-2 snow depth NMAD and R2 compared to the airborne lidar snow depth by smoothing length. Will add the following text to the results around L329:

"Comparing ICESat-2 snow depths against Mores Creek airborne lidar snow depth shows that when comparing data with the same spatial coverage, ICESat-2 snow depth uncertainty and correlation both improve with smoothing length. The ICESat-2 uncertainty drops ~0.25 m across all smoothing lengths while R2 rises from ~0.48 to ~0.60. R2 is above 0.5 for all smoothing lengths > 300 m."

Comparison of MSC ICESat-2 snow depth and Helicopter lidar draft figure:

[Figure]

Table 4 figure replacement draft:

[Figure]

L360: Could it also be related to the fact that applying time-variant co-registration reduces the number of samples available and biases the dataset toward specific slope/topographical regions, increasing the noise in the registration? Maybe looking at the number of return photons can help reduce this issue by reducing the impact of mixed surface types where snow and snow-free terrain overlap.

It is definitely possible that the individual co-registrations are biased because they fall in areas with certain attributes that are not representative of the broader area. We will add text to L359-263 to highlight this point. Additions in bold.

Proposed rewrite: "Most concerningly, the application of a time-variant co-registration transform resulted in no correlation between ICESat-2 snow depths and precise independent snow depth estimates (Table 2). **The high variability in time-variant co-registration of sets may be due to the limited spatial coverage of an individual overpass. If a given overpass happens to fall over a highly sloped or densely vegetated area the increased uncertainty or bias will impact the accuracy of the co-registration.** The poor performance of individual co-registration transforms **in the winter** is likely **also** due to sparse

snow-free winter terrain. Snow coverage obscures stable terrain and when <10% of the region of interest is stable the accuracy of co-registration decreases with the percent of stable terrain (Nuth and Kääb, 2011)."

L416: How large are these negative values? To reduce the risk of biasing the snow depth when removing SD < 0, could you allow for smaller negative values to be kept, perhaps within some limit or error?

The median negative snow depth is ~ -0.7 m with an interquartile range of ~1.1 m. We tested setting the negative threshold at -0.3 m instead of 0 m based on the median bias calculated from comparing ICESat-2 and airborne derived snow depths at Mores Creek, however we observed an increase in uncertainty and bias in the ICESat-2 snow depth data. We propose to add the following text after L416 to expand on this.

Proposed addition: "The overestimation of shallow snow depths could be mitigated by lowering the minimum snow depth threshold below 0 m based on ICESat-2 snow depth uncertainty, however this also increases the impact of outliers on deeper snow depth estimates."

L430: I think grouping them into elevation zones rather than horizontal distance bins would be a more effective approach, as you will increase data density. That's why I suggested calculating the spatial autocorrelation: you can use that to first get all data within that distance and then group them in elevation bands.

We agree that grouping data into elevation zones is likely a more effective approach as we can include a larger amount of data while maintaining a relatively high spatial resolution. In line with what we expect from forested mountain environments given that snow depth correlation length is typically much shorter than the 20 m ATL06_SR segment length in such environments (Trujillo et al., 2009), the auto correlation is greatest without lag and falls away precipitously as the spatial lag increases. Thus the smoothing scale must be a compromise between keeping a high spatial resolution and including sufficient data. We propose to add the following after L431 to expand the discussion of elevation zones:

"Grouping the ICESat-2 snow depth data into elevation zones may be more effective for characterizing snow depths across a landscape than directly calculating spatial distribution as it maintains a high spatial resolution while averaging over many data points to reduce variability. This assumes that the primary control on snow depth is elevation; generally this applies (fig 7), however slope and vegetation density varies greatly across this terrain therefore there can be large variation in snow depth at a given elevation. Grouping the data by elevation zones may obscure other terrain controls on snow depth. However terrain controls on snow depth will also be obscured by estimating snow depth at the larger spatial smoothing scales required to achieve similar data density."

Trujillo, E., Ramírez, J. A., and Elder, K. J.: Scaling properties and spatial organization of snow depth fields in sub-alpine forest and alpine tundra, Hydrol. Process., 23, 1575–1590, https://doi.org/10.1002/hyp.7270, 2009.